# Neuropeptidergic circuit modulation of developmental sleep in *Drosophila*

Chikayo Hemmi[1], Kenichi Ishii[1]*, Mana Motoyoshi[2], Masato Tsuji[1], Kazuo Emoto[1,2]*

[1]Department of Biological Sciences, The University of Tokyo, Tokyo, Japan; [2]International Research Center for Neurointelligence (WPI-IRCN), The University of Tokyo, Tokyo, Japan

## eLife Assessment

The study investigates an emerging research field: the interaction between sleep and development. The authors used Drosophila larvae sleep as a study model and provide insight into how neuropeptide circuitry control sleep differentially between larvae and adult Drosophila. By using board range of behaviour and imaging methods and analysis, the authors provide a **valuable** investigation that demonstrates a larvae-specific sleep regulatory neural pathway of Hugin-PK2-Dilps in the Drosophila neurosecretory centre IPC. While some further text clarifications are still required, the revision presented **convincing** evidence supporting the claims with the new imaging data, sleep parametric analysis, and further clarification addressing the reviewers' comments.

**\*For correspondence:**
ken1ishii2@gmail.com (KI);
emoto@bs.s.u-tokyo.ac.jp (KE)

**Competing interest:** The authors declare that no competing interests exist.

**Abstract** Sleep–wakefulness regulation dynamically evolves along development in a wide range of organisms. While the mechanism regulating sleep in adults is relatively well understood, little is known about its counterpart in early developmental stages. Here, we report a neuropeptidergic circuitry that modulates sleep in developing *Drosophila* larvae. Through an unbiased screen, we identified the neuropeptide Hugin and its receptor PK2-R1 as critical regulators of larval sleep. Our genetic and behavioral data suggest that HugPC neurons secrete Hugin peptides to activate insulin-producing cells (IPCs), which express a Hugin receptor PK2-R1. IPCs, in turn, release insulin-like peptides (Dilps) to regulate sleep. We further show that the Hugin/PK2-R1 axis is dispensable for adult sleep. Our findings thus reveal the neuromodulatory circuit that regulates developmental sleep in larvae and highlight differential impacts of the same modulatory axis on early-life sleep and adult sleep.

## Introduction

Sleep is a fundamental state with significant impacts on many aspects of cognition and metabolism (*Blumberg et al., 2005*; *Davis et al., 2004*; *Sorribes et al., 2013*). While sleep remains essential throughout life, its regulation dynamically unfolds over the course of development. For instance, the pace of sleep–wakefulness cycle is relatively fast in infants, while gradually slowing down and stabilizing in adults (*Blumberg et al., 2005*; *Davis et al., 2004*; *Sorribes et al., 2013*). Furthermore, the sleep–wake cycle in infants is typically independent of circadian rhythm, but it gradually comes under the circadian regulation (*Davis et al., 2004*; *Frank et al., 2017*; *Poe et al., 2023*). Such developmental evolution of sleep–wakefulness cycles appears to be conserved across a wide range of organisms including flies, fishes, mammals, and humans (*Blumberg et al., 2005*; *Davis et al., 2004*; *Poe et al., 2023*; *Sorribes et al., 2013*; *Szuperak et al., 2018*). In contrast to deep insights into the molecular and neural mechanisms underlying adult sleep regulation (*Barlow and Rihel, 2017*; *Scammell et al., 2017*; *Shafer and Keene, 2021*), little is known about regulatory mechanisms underlying

developmental sleep in infants, partially due to a lack of convenient models to study developmental sleep.

*Drosophila* larvae have recently emerged as a suitable model to study sleep regulation mechanisms. Recent studies have reported that the second instar larvae show short periods (<6 s) of 'inactive state' during locomotion. Importantly, such 'inactive state' is consistent with the general definition of sleep: reduced responsiveness to noxious stimuli, homeostatic response to sleep deprivation, and rapid reversibility upon stimulation (*Szuperak et al., 2018*), suggesting that the 'inactive state' in larvae likely corresponds to the sleep state. Interestingly, loss-of-function mutations in the clock genes *clock* and *cyc* fail to impact the larval sleep, while significantly altering the sleep patterns in adult flies (*Dubowy and Sehgal, 2017*; *Hendricks et al., 2003*; *Szuperak et al., 2018*). Likewise, mutations in the dopamine transporter *DAT* fail to influence larval sleep, while reducing sleep amounts in adult (*Kume et al., 2005*; *Szuperak et al., 2018*). These observations imply that sleep regulation mechanisms might be at least in part distinct between larvae and adults, yet the neural mechanism of larval sleep remains vastly understudied compared to that of adult sleep.

In this study, we performed an unbiased genetic screen in the second instar larvae and identified the neuropeptide Hugin and its receptor PK2-R1 as a pair critical for larval sleep. At the circuit level, Hugin-producing HugPC neurons directly stimulate the insulin-producing cells (IPCs) via PK2-R1 to regulate sleep. We further found that regulators of larval sleep are either irrelevant or exert the opposite effects on adult sleep. Overall, our findings uncover the neuropeptidergic circuitry that regulates developmental sleep in larvae and highlight mechanistic differences between larval and adult sleep.

## Results and discussion
### Results PK2-R1 is required for wake/sleep control in *Drosophila* larvae

To understand molecular and neural mechanisms of larval sleep, we modified a previously reported system (*Churgin et al., 2019*; *Szuperak et al., 2018*) to automatically quantify the sleep amount of second instar larvae (*Figure 1—figure supplement 1A–C*) (see Materials and methods for details).

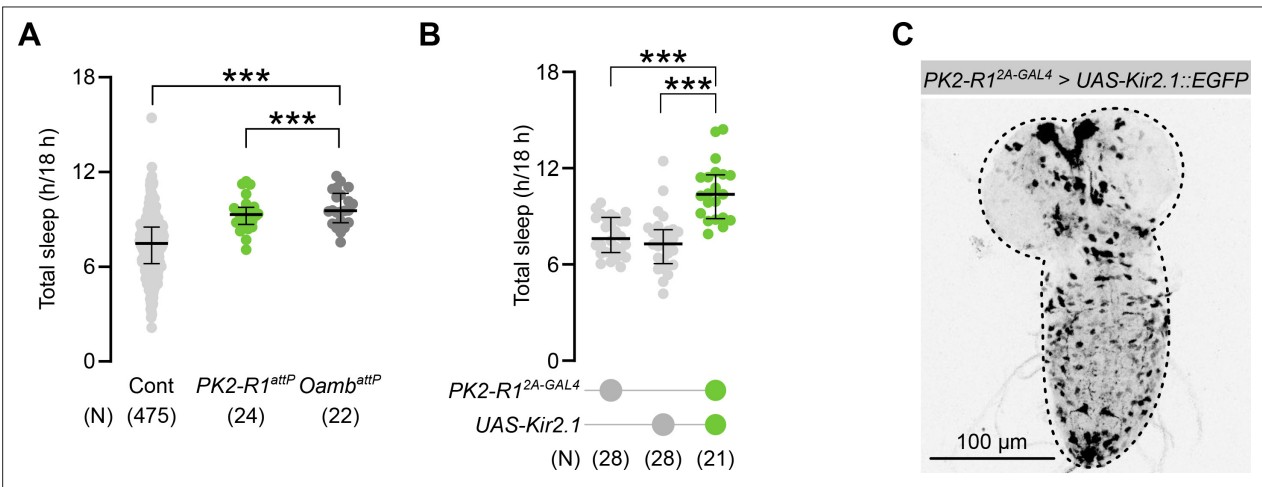

**Figure 1.** PK2-R1 is required for larval sleep control. (**A**) Sleep amounts in *PK2-R1* or *Oamb* knockout mutants. Each dot represents an individual animal; in this and the following panels, '*N*' indicates the number of biologically independent animals per group, and the thick line and thin error bars indicate the median and interquartile range (Q1–Q3), respectively. ***p < 0.001 (Mann–Whitney *U*-test with Bonferroni correction). (**B**) Sleep amounts in larvae in which *PK2-R1* neurons were silenced. ***p < 0.001 (Mann–Whitney *U*-test with Bonferroni correction). (**C**) Expression pattern of *PK2-R1-GAL4 > UAS-Kir::EGFP* larvae.

The online version of this article includes the following figure supplement(s) for figure 1:

**Figure supplement 1.** Detailed experimental setup for analyzing larval sleep.

**Figure supplement 2.** Larval 'sleep state' in this study is consistent with behavioral criteria for sleep.

**Figure supplement 3.** CRISPR-knockout screen for genes that regulate larval sleep.

**Figure supplement 4.** Effects of neuronal manipulation on larval sleep.

**Figure supplement 5.** Larval sleep phenotypes in PK2-R1 mutants quantified over 18 hr and the first 6 hr.

Our system automatically annotates larvae in each frame as either 'inactive' or 'active' states on the basis of pixel changes, with ~90% accuracy compared to manual annotation (*Figure 1—figure supplement 1C*; see Methods for details). We found that bouts of ≥12 inactive frames are consistent with the general criteria for sleep; reduced responsiveness to noxious stimuli, homeostatic response to sleep loss, and rapid reversibility upon stimulation (*Raizen et al., 2008*; *Szuperak et al., 2018*; *Figure 1—figure supplement 2B–D*). Therefore, the present study defines sleep as an inactive state of ≥12 consecutive frames (*Figure 1—figure supplement 1A*).

Using this system, we next carried out an unbiased screen for genes involved in sleep regulation. To this end, we tested 47 CRISPR-knockout lines of genes encoding enzymes or receptors for neuropeptides and monoamines (*Deng et al., 2019*; *Supplementary file 1*). As illustrated in *Figure 1—figure supplements 3*, 13 out of 47 homozygous null mutants exhibited significant changes in sleep amounts compared to the control group. To further test the functions of these candidate genes, we next took advantage of the CRISPR-knock-in drivers in which GAL4 sequence is inserted into each gene locus. With these drivers, we silenced neurons expressing each candidate gene by expressing the inward-rectifying potassium channel Kir2.1 (*Baines et al., 2001*). Silencing neurons expressing *Oamb* and *PK2-R1* resulted in significant sleep increase, phenocopying the knockout of these genes (*Figure 1A, B*, *Figure 1—figure supplement 4D*). *Oamb* encodes the octopamine receptor and has been reported to be involved in larval sleep (*Szuperak et al., 2018*). The role of PK2-R1 in larval sleep, on the other hand, has been unknown to date. Given its strong expression in IPCs (*Schlegel et al., 2016*) and its function as a receptor for the neuropeptide Hugin, which modulates feeding (*Schoofs et al., 2014*), we hypothesized that PK2-R1 might mediate neuropeptidergic signaling that links metabolic and sleep regulation during development. We thus focused on this gene as a candidate connecting behavioral and endocrine sleep control.

## IPCs express PK2-R1 and regulate sleep states in developing larvae

Since the *PK2-R1* driver broadly labeled neurons (~1000 cells) throughout the CNS (*Figure 1C*), we next attempted to identify a subpopulation of these neurons relevant for sleep regulation. To narrow down the candidates within PK2-R1-positive populations, we examined subsets labeling GABAergic, cholinergic, or glutamatergic neurons, as well as those co-expressing other neuropeptides identified in our screen, but none of these manipulations reproduced the sleep phenotype caused by PK2-R1 perturbation. Notably, IPCs in the pars intercerebralis, the neurosecretory center of *Drosophila*, have been reported to express *PK2-R1* (*Schlegel et al., 2016*). Consistently, the *PK2-R1²ᴬ⁻ᴸᵉˣᴬ* knock-in driver labeled all IPCs (*Figure 2A, B*). To assess whether IPCs are involved in larval sleep, we silenced IPCs with Kir2.1 expression, using the IPC-specific drivers *Dilp3-* and *Dilp5-GAL4* (*Figure 2C*). We found that silencing of IPCs significantly increased the sleep amounts compared to the control, phenocopying *PK2-R1* neuron silencing and *PK2-R1* knockout (*Figure 1A*). Consistently, both *Dilp3* and *Dilp5* null mutants exhibited larger sleep amounts compared to the control (*Figure 2D*). These phenotypes of *Dilp* mutants and IPC silencing are unlikely to be accounted for by locomotion defects, as the travel distances during wake periods were unaffected (*Figure 2—figure supplement 1*). Through these trials, IPCs emerged as one of the populations whose silencing phenocopied PK2-R1 loss, supporting their key role in larval sleep regulation while not excluding contributions from other PK2-R1-expressing neurons. Collectively, these findings suggest that *PK2-R1* in IPCs regulates larval sleep.

## Hugin-expressing neurons control larval sleep states

Given that PK2-R1 is a receptor of the neuropeptide Hugin (*Rosenkilde et al., 2003*), we next examined whether Hugin is involved in the larval sleep. We found that *Hug* mutant larvae exhibited significantly larger sleep amounts compared to the control (*Figure 3A*), consistent with *PK2-R1* knockouts (*Figure 1A*). When we silenced Hug-expressing neurons by driving the expression of Kir2.1 with the *Hug* CRISPR-knock-in driver *Hug²ᴬ⁻ᴳᴬᴸ⁴* (*Deng et al., 2019*), it resulted in a significant increase of sleep amounts without detectable changes in locomotion speed (*Figure 3B*, *Figure 3—figure supplement 1B*). Conversely, activation of *Hug*-expressing neurons with the red-shifted channel rhodopsin ReaChR (*Inagaki et al., 2014*) or heat-sensitive ion channel TrpA1 (*Hamada et al., 2008*) both caused a significant reduction of sleep amount (*Figure 3C, D*) and increased locomotor activity. However, locomotion changes were not consistently observed upon either activation or suppression of Hug neurons (*Figure 3—figure supplement 1*), suggesting that changes in sleep cannot be simply explained by

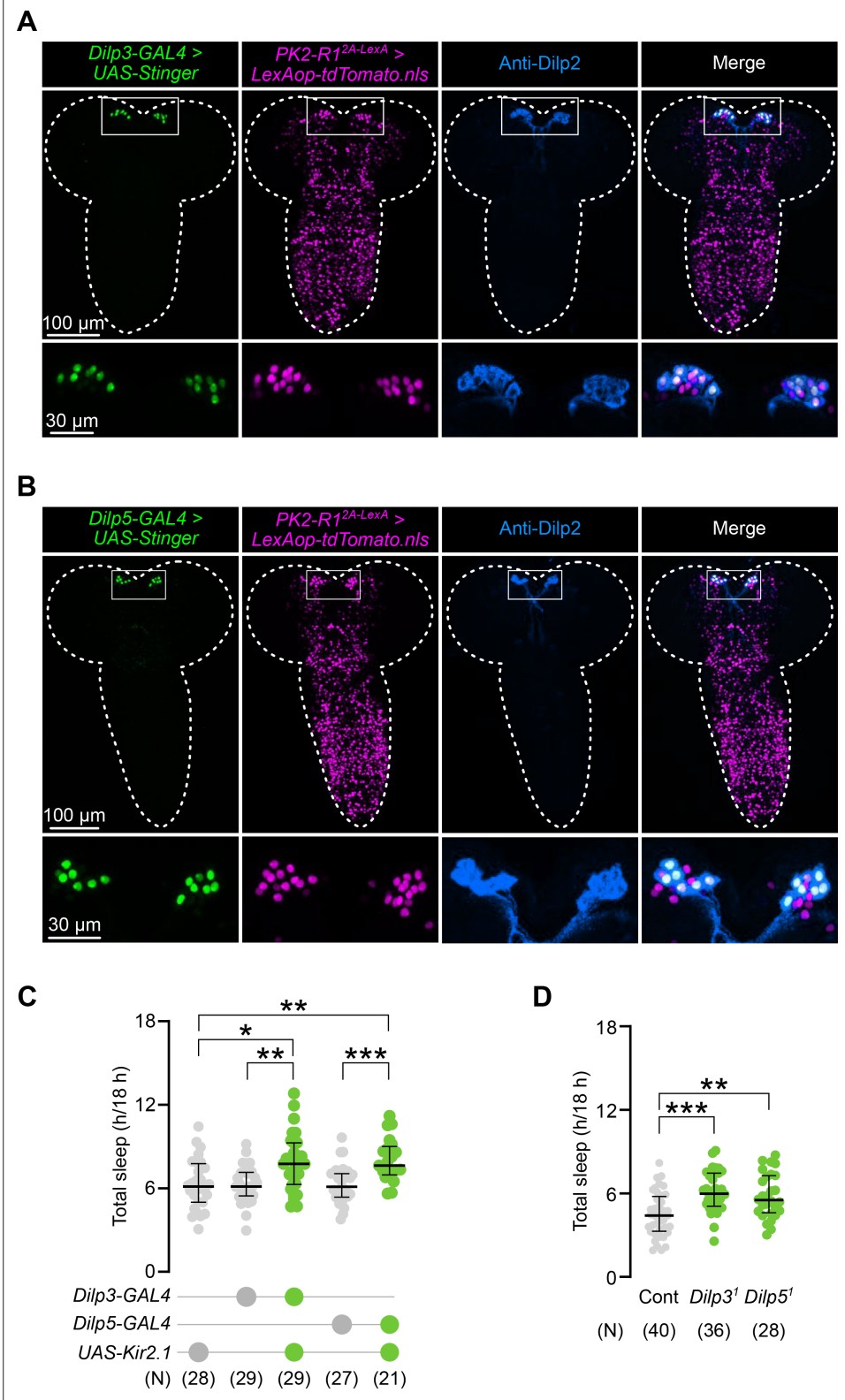

**Figure 2.** Insulin-producing cells (IPCs) and Dilps negatively regulate larval sleep. (**A**) Triple labeling of *PK2-R1* neurons expressing nuclear-localized RFP (magenta), *Dilp3* neurons expressing nuclear-localized GFP (green), and anti-Dilp2-positive cells (blue). Top panels show signals in the larval brain and the VNC. Bottom panels show magnified images of the white-squared area in each top panel, focusing on the dorsomedial brain region

*Figure 2 continued on next page*

*Figure 2 continued*

where the cell bodies of IPCs are located. Note that all IPCs labeled with *Dilp3-GAL4* overlapped with *PK2-R1^{2A-LexA}*-expressing cells. Similar results were obtained from five independent samples of the same genotype. (**B**) Simultaneous detection of *PK2-R1* neurons expressing nuclear-localized RFP (magenta), *Dilp5* neurons expressing nuclear-localized GFP (green), and anti-Dilp2-positive cells (blue). Similar results were obtained from five independent samples of the same genotype. (**C**) Effect of IPC silencing on larval sleep. Each dot represents an individual animal; in this and the following panels, '*N*' indicates the number of biologically independent animals per group, and the thick line and thin error bars indicate the median and interquartile range (Q1–Q3), respectively. ***p < 0.001, **p < 0.01, *p < 0.05 (Mann–Whitney *U*-test with Bonferroni correction). (**D**) Sleep amounts in *Dilp3* or *Dilp5* null mutants. ***p < 0.001, **p < 0.01 (Mann–Whitney *U*-test with Bonferroni correction).

The online version of this article includes the following figure supplement(s) for figure 2:

**Figure supplement 1.** Locomotion speeds are not consistently affected by genetical manipulations of *PK2-R1*, insulin-producing cells (IPCs), or *Dilps*.

locomotor alterations. We further quantified larval feeding using a dye-based ingestion assay and found that silencing HugPC neurons reduced food intake (*Figure 3—figure supplement 3*), indicating that the sleep phenotype is unlikely to be explained by feeding behavior being misclassified as sleep. These data together indicate that *Hug*-expressing neurons are responsible for downregulating larval sleep.

*Hug*-expressing neurons are comprised of 20 cells in the subesophageal zone of the brain and are classified into four subpopulations based on their projection patterns (*Bader et al., 2007*). Of these, one subpopulation called HugPC neurons *Bader et al., 2007* have been reported to form synaptic connections with IPCs (*Hückesfeld et al., 2016*). We thus suspected that HugPC neurons may be responsible for sleep regulation. In line with this notion, axons of HugPC neurons, labeled by a newly generated *HugPC-LexA* driver (*Figure 3—figure supplement 2*), and IPCs, labeled by *Dilp3-GAL4* (*Figure 3E*), exist in space close to each other. Indeed, silencing of HugPC neurons by Kir2.1 significantly increased larval sleep (*Figure 3F*), phenocopying those observed in *Hugin* and *PK2-R1* mutants. These results hint at the possibility that HugPC neurons activate IPCs via Hugin/PK2-R1 signaling to regulate sleep.

## Hugin triggers Ca$^{2+}$ elevation and Dilp secretion in larval IPCs via PK2-R1

A previous report showed that IPCs exhibit increased firing rates and intracellular calcium (Ca$^{2+}$) levels upon stimulation (*Kréneisz et al., 2010*). This motivated us to next test if Hugin activates IPCs. To further visualize the output pattern of HuginPC neurons, we expressed the presynaptic marker Syt-eGFP in HuginPC neurons (*Figure 4—figure supplement 1*). To test whether Hugin activates IPCs, we utilized the recently developed calcium indicator CRTC::GFP, which can report Ca$^{2+}$ dynamics in the timescale of several minutes (*Bonheur et al., 2023*). As glucose reportedly triggers Ca$^{2+}$ dynamics in IPCs in a range of minutes (*Oh et al., 2019*), we wondered if CRTC::GFP can detect this dynamics, as a positive control. To this end, we applied D-glucose to isolated larval brains and found a significant cytosol-to-nuclear translocation of CRTC::GFP in IPCs within ~5 min, confirming that this reporter can indeed detect Ca$^{2+}$ responses in IPCs upon stimulation (Figure 5B). We then examined whether *Hug* neurons act upstream of *PK2-R1*-expressing IPCs by measuring Ca$^{2+}$ responses of IPCs while thermogenetically activating *Hug* neurons with TrpA1. We found that activation of *Hug* neurons tended to increase Ca$^{2+}$ levels in IPCs (*Figure 4B, C*), consistent with the HuginPC-to-IPCs axis (*Figure 4D*). Next, we tested whether Hug peptides can induce Ca$^{2+}$ responses in IPCs using ex vivo preparations. The *Hug* gene encodes a precursor that potentially produces at least two distinct peptides, Hug-γ and PK-2, both of which can bind PK2-R1 to induce Ca$^{2+}$ responses (*Rosenkilde et al., 2003*). We thus incubated the CRTC::GFP-expressing larval brain with chemically synthesized Hug-γ or PK-2 (*Kréneisz et al., 2010*; *Oh et al., 2019*; *Figure 5A*). We found that bath application of either Hug-γ or PK-2 induced significant Ca$^{2+}$ responses in IPCs (*Figure 5C, D*). In agreement with the idea that Hug peptides activate IPCs via *PK2-R1*, knockout of *PK2-R1* blocked the Hug peptide-evoked Ca$^{2+}$ elevations in IPCs (*Figure 5F, G*). Similar results were obtained when we bath-applied the synthetic Neuromedin U (NMU), the mammalian homolog of Hugin (*Figure 5A, C, G*), suggesting the role of Hugin/PK2-R1 signaling may be conserved across species. Furthermore, Dilp3 accumulated in IPCs

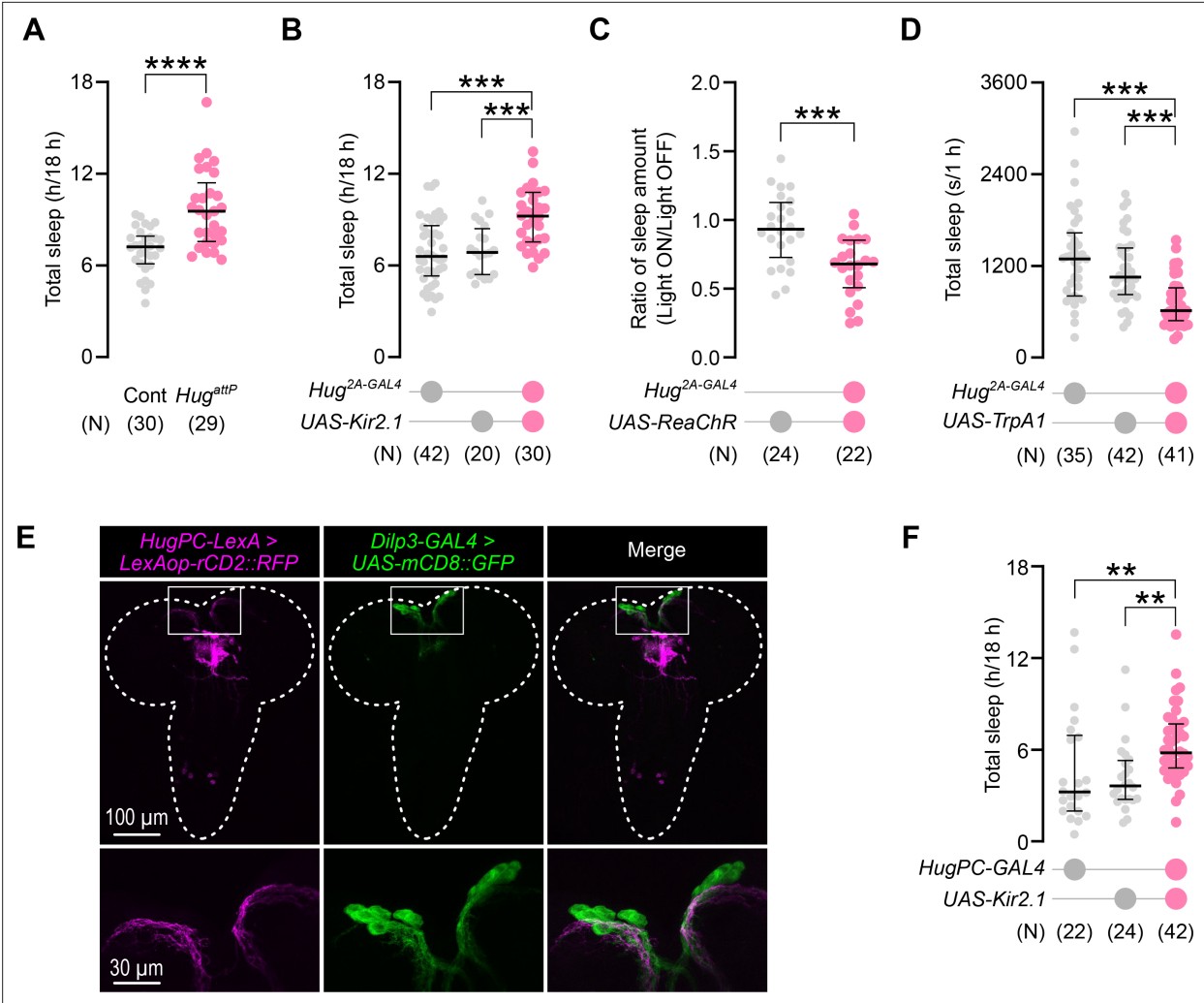

**Figure 3.** *Hug* neurons negatively regulate larval sleep. (**A**) Sleep amounts in *Hug* CRISPR-knockout mutant larvae. ****p < 0.0001 (Mann–Whitney *U*-test). Each dot represents an individual animal; in this and the following panels, '*N*' indicates the number of biologically independent animals per group, and the thick line and thin error bars indicate the median and interquartile range (Q1–Q3), respectively. (**B**) Effect of silencing Hug neurons on larval sleep amount. **p < 0.01, *p < 0.05 (Mann–Whitney *U*-test with Bonferroni correction). (**C**) Larval sleep changes induced by optogenetic activation of *Hug* neurons. For each genotype, sleep duration in the 1 hr light-ON period was normalized to that in the 1 hr light-OFF phase. *p < 0.05 (Mann–Whitney *U*-test with Bonferroni correction). (**D**) Larval sleep during thermogenetic activation of *Hug* neurons. **p < 0.01, *p < 0.05 (Mann–Whitney *U*-test with Bonferroni correction). (**E**) Visualization of HugPC neurons and insulin-producing cells (IPCs) labeled by rCD2::RFP (magenta) and mCD8::GFP (green), respectively. The bottom panels show magnified images of the white-squared dorsomedial region in the top panels, where HugPC neurons project their axons close to the cell bodies of the IPCs. While the top panels are z-stacks of 122 image slices (1 μm interval) covering the entire brain tissue, the bottom panels are projections of 60 slices centering around IPCs. Similar results were obtained from five independent samples of the same genotype. (**F**) The effect of neuronal silencing confined to the HugPC subpopulation. **p < 0.01, *p < 0.05 (Mann–Whitney *U*-test with Bonferroni correction).

The online version of this article includes the following figure supplement(s) for figure 3:

**Figure supplement 1.** Effects of Hug pathway manipulations on larval locomotion speed.

**Figure supplement 2.** Generation of *HugPC-LexA* transgenic lines.

**Figure supplement 3.** Silencing HugPC neurons reduces larval food intake measured by a dye-based assay.

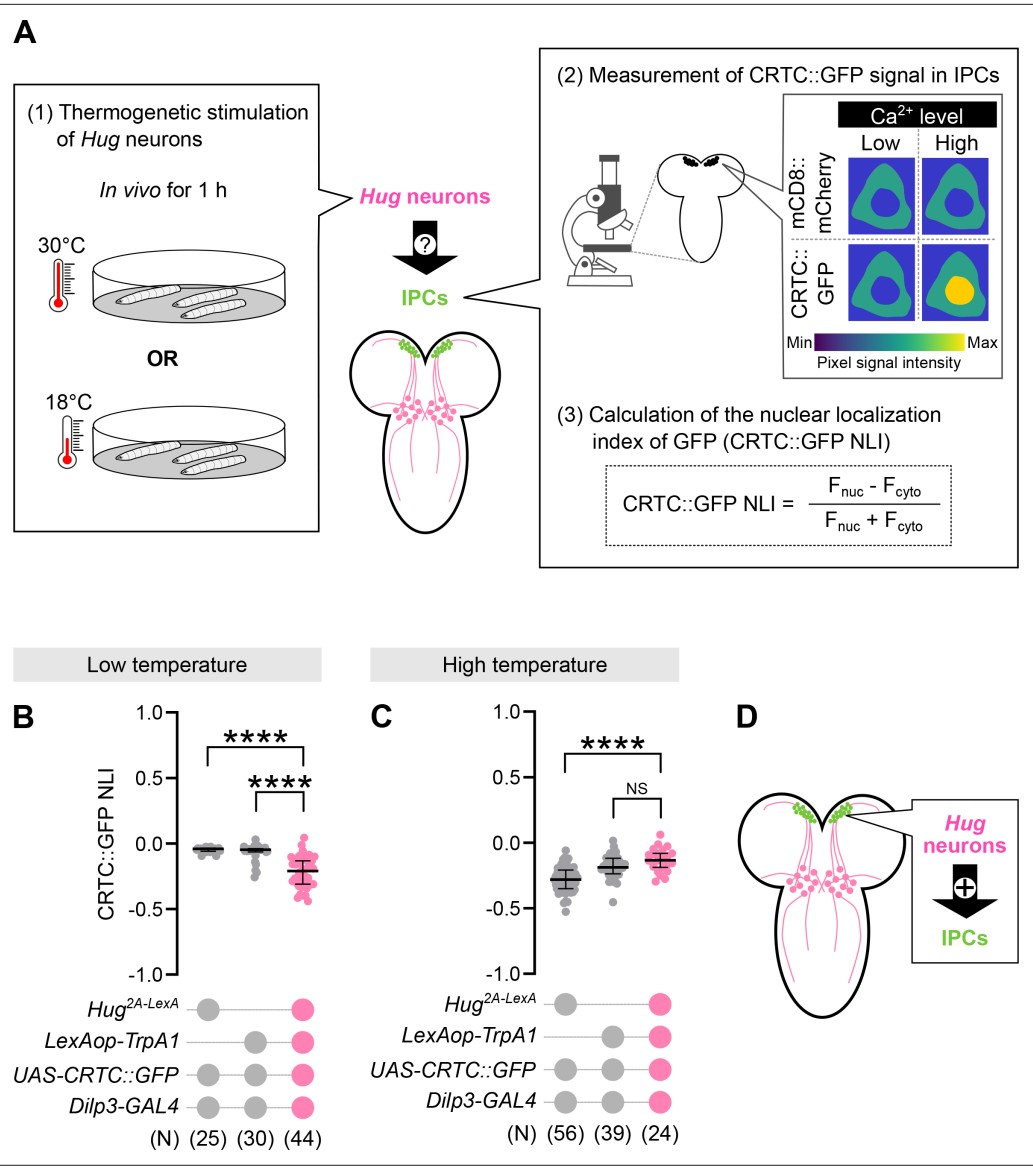

**Figure 4.** Activation of Hug neurons triggers Ca²⁺ responses in larval insulin-producing cells (IPCs). (**A**) Schematic workflow for assessing intracellular Ca²⁺ levels in IPCs while thermogenetically activating Hug neurons. Larvae were exposed to low (18°C) or high (30°C) temperature for 1 hr, followed by imaging of CRTC::GFP in IPCs and calculation of the CRTC::GFP nuclear localization index (NLI; see Methods). Nuclear-localized CRTC::GFP signal (CRTC::GFP NLI) in larval IPCs under low-temperature (**B**) or high-temperature (**C**) conditions, with or without thermogenetic activation of Hug neurons. Each dot represents an individual cell; the thick line indicates the median and the thin error bars indicate the interquartile range (Q1–Q3). '*N*' indicates the number of cells analyzed per group. ****p < 0.0001, NS: p ≥ 0.05 (Mann–Whitney *U*-test). (**D**) Working model of a neuronal network in which Hug neurons activate IPCs to regulate larval sleep.

The online version of this article includes the following figure supplement(s) for figure 4:

**Figure supplement 1.** Axonal projections of HuginPC neurons visualized with a presynaptic marker.

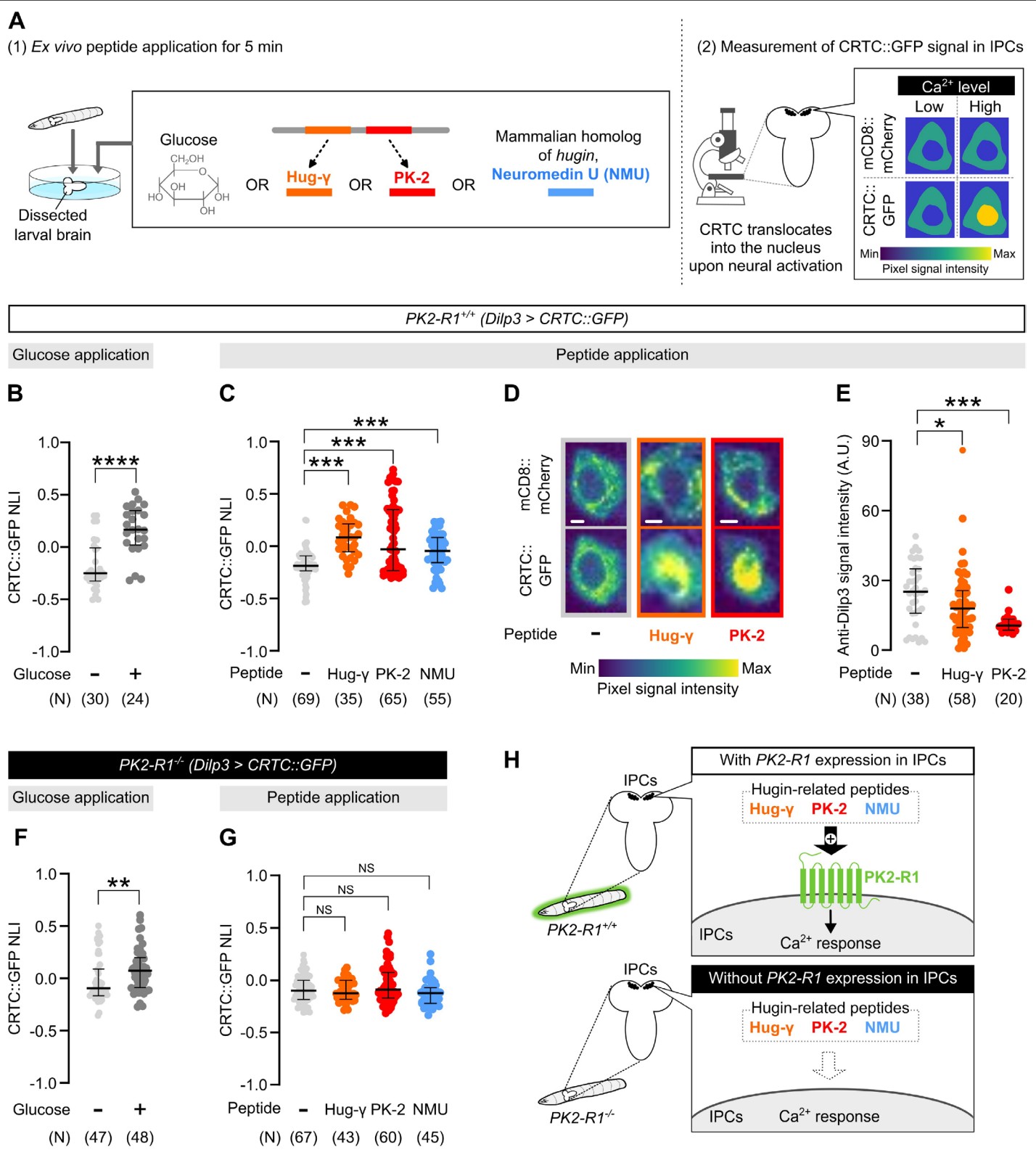

**Figure 5.** Hug peptides induce Ca²⁺ responses in larval insulin-producing cells (IPCs) via PK2-R1. (**A**) Schematic flow of peptide application followed by Ca²⁺ imaging. Ca²⁺ responses in larval IPCs during ex vivo bath application of either glucose (**B**) or Hug peptides (**C**). In this and the following panels, '*N*' indicates the number of cells used for each group, and the thick line and thin error bar represent the median and interquartile range (Q1–Q3), respectively. ****p < 0.0001, ***p < 0.001 (Mann–Whitney *U*-test with Bonferroni correction). (**D**) Representative images of larval IPCs upon peptide

*Figure 5 continued on next page*

*Figure 5 continued*

application. Scale bars, 2 μm. (**E**) Anti-Dilp3 signal intensity measured within the cytosolic areas of larval IPCs. ***p < 0.001, *p < 0.05 (Mann–Whitney *U*-test with Bonferroni correction). Ca$^{2+}$ responses in larval IPCs derived from *PK2-R1* knockout mutants, measured after either glucose (**F**) or Hug peptide application (**G**). **p < 0.01, NS: p ≥ 0.05 (Mann–Whitney *U*-test with Bonferroni correction). (**H**) A model where Hug peptides, but not glucose, activate IPCs via the PK2-R1 receptor in the larval brain.

The online version of this article includes the following figure supplement(s) for figure 5:

**Figure supplement 1.** D-glucose but not Hug peptide causes Dilp2 reduction in larval insulin-producing cells (IPCs).

**Figure supplement 2.** Representative images of Dilp3 immunoreactivity in larval insulin-producing cells (IPCs) after bath application of Hug-γ peptide.

was significantly reduced after applying Hug peptides, suggesting that Hugin causes IPCs to release Dilp3 (*Figure 5E*). Collectively, these data suggest that *Hug* neurons activate IPCs via the Hug/PK2-R1 signaling to regulate sleep (*Figure 5H*).

## Hugin/PK2-R1 axis has distinct impacts on wake/sleep control in larvae and adults

Given the differences between larval vs adult sleep in temporal pattern and circadian influence, we next wondered if Hug/PK2-R1/Dilps axis functions in adult sleep as well. As in larvae, *PK2-R1* knockout adults showed increased sleep amount compared to the control (*Figure 6—figure supplement 1B*). In contrast, *Hug* knockout or silencing of HugPC neurons failed to influence adult sleep (*Figure 6A, B*). This suggests that, unlike larvae, Hug is dispensable for adult sleep. Interestingly, we observed that the expression patterns of PK2-R1 and Hug, and the morphology of HugPC neurons and IPCs, showed a broadly similar distribution in larvae and adults, although we did not directly track individual neurons across development (*Figure 6—figure supplement 2*). This implies that the differential roles of Hug in larvae vs adults are likely due to physiological differences in HugPC neurons and/or IPCs. We thus examined how Hug peptides evoke Ca$^{2+}$ responses in adult IPCs and found that neither Hug-γ nor PK-2 could induce Ca$^{2+}$ responses in adult IPCs (*Figure 6D, E*), unlike in larvae. Although PK-2 treatment even led to a modest increase in Dilp3 immunoreactivity in adult IPCs (*Figure 6F*), the physiological significance of this effect remains unclear; at present, we consider it most likely that PK-2 acts on larval and adult IPCs in a different manner. These data indicate that Hug induces Ca$^{2+}$ responses and Dilps secretion in larval IPCs but not in adult IPCs. Surprisingly, we further found that *Dilp3* or *Dilp5* null mutations reduced sleep amounts in adults, consistent with previous reports (*Cong et al., 2015*; *Figure 6C*), while in larvae, these mutations in fact increased the sleep amounts. These data unexpectedly reveal how the same set of neuronal circuits and molecular signaling therein can evolve over development in a complex manner.

## Discussion

In this study, we have identified the neuropeptide Hugin and its receptor PK2-R1 as a ligand/receptor pair critical for sleep regulation in *Drosophila* larvae. We have further shown that larval IPCs express PK2-R1, respond, and release Dilp3 upon Hugin stimulation. Surprisingly, the Hugin/PK2-R1 axis is dispensable for sleep regulation in adults, even though the gene expressions as well as the circuit structure appeared to be conserved between larval and adult brains. Furthermore, Dilps appear to modulate sleep in opposite directions in larvae and adults. Our findings thus uncover the neuropeptidergic modulation circuitry that specifically regulates developmental sleep and suggest divergent usage of the same molecules and circuitry for sleep modulation in larvae and adults.

### Hugin/PK2-R1/Dilps axis negatively regulates sleep in larvae

The present study shows that the neuropeptide Hugin and its receptor PK2-R1 are required for sleep regulation in the early larval stage. This notion is supported by the following lines of evidence. First, knockout mutations in *Hugin* or *PK2-R1* increased sleep amounts compared to the control (*Figures 1A and 3A*). Consistently, silencing a subpopulation of *Hugin*-expressing neurons, called HugPC neurons, or *PK2-R1* neurons increased sleep amounts (*Figures 1B and 3F*). Second, thermogenetic activation of Hugin neurons decreased sleep amounts (*Figure 3D*). Third, HugPC neurons project their axons to IPCs in the brain that express PK2-R1 receptors (*Figure 3E*). Fourth, Hugin peptides induced Ca$^{2+}$

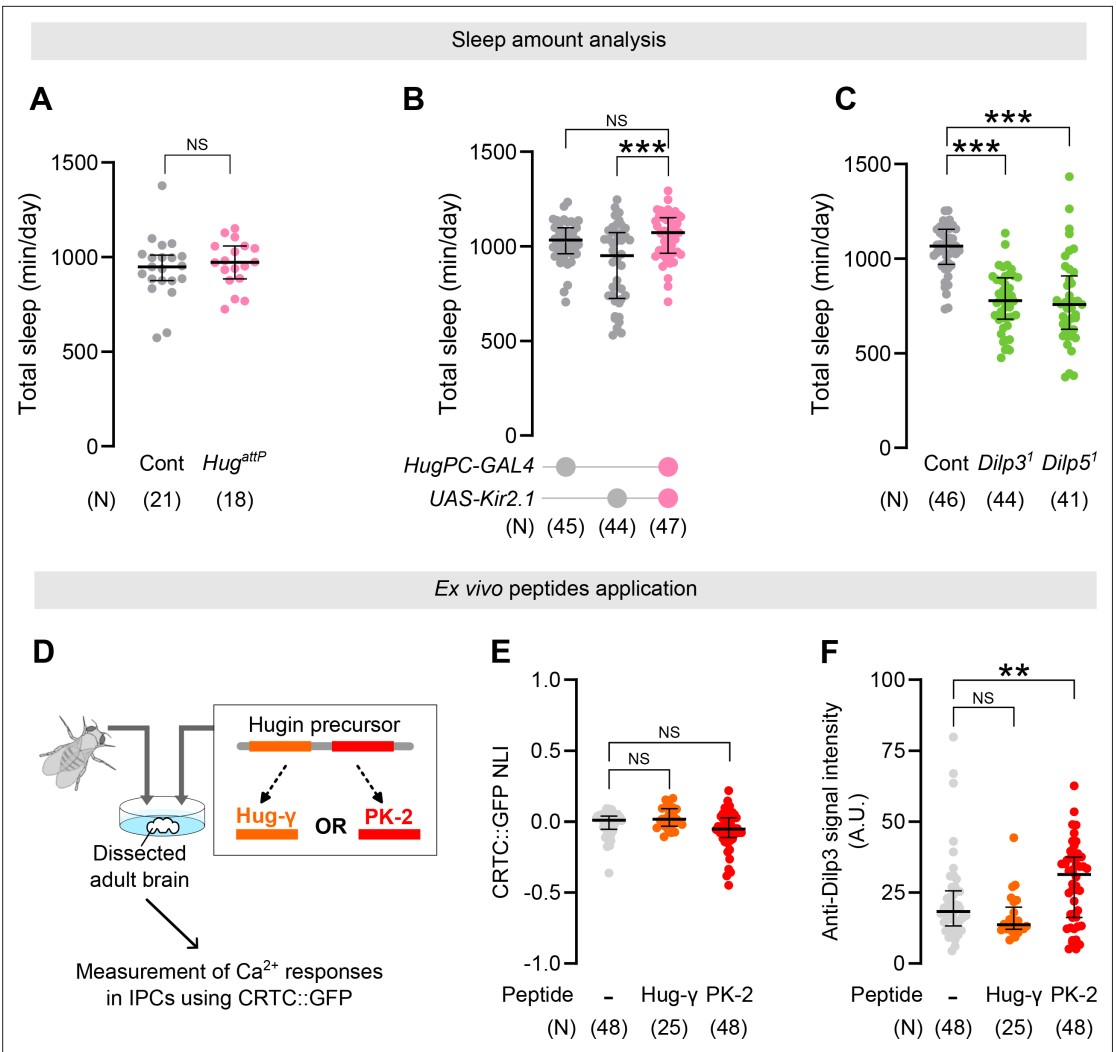

**Figure 6.** Distinct impacts of the Hugin/PK2-R1 axis on wake/sleep control in larvae and adult. (**A**) Total sleep amounts in *Hug knockout* mutant adults. NS: p ≥ 0.05 (Mann–Whitney *U*-test). In these and the following panels, '*N*' indicates the number of biologically independent animals used for each group, and the thick line and thin error bar represent the median and interquartile range (Q1–Q3), respectively. NS p ≥ 0.05 (Mann–Whitney *U*-test). (**B**) Sleep amounts of adults in which HugPC neurons were silenced. ***p < 0.001, NS: p ≥ 0.05 (Mann–Whitney *U*-test with Bonferroni correction). (**C**) Sleep amounts of *Dilp3* or *Dilp5* null mutant adults. ***p < 0.001 (Mann–Whitney *U*-test with Bonferroni correction). (**D**) Schematic flow of peptide application experiments using adult brains. Ca²⁺ responses (**E**) or anti-Dilp3 signal intensity (**F**) in adult insulin-producing cells (IPCs) after Hug peptide application. **p < 0.01, NS: p ≥ 0.05 (Mann–Whitney *U*-test with Bonferroni correction).

The online version of this article includes the following figure supplement(s) for figure 6:

**Figure supplement 1.** Sleep patterns of adult flies with genetic manipulations in the Hugin and insulin pathways.

**Figure supplement 2.** Morphologies of *PK2-R1* neurons, HugPC neurons, and insulin-producing cells (IPCs) in the adult brain.

**Figure supplement 3.** Adult sleep architecture and waking activity following genetic manipulations of the Hugin and insulin pathways.

elevation and promoted Dilp3 release from IPCs in a PK2-R1-dependent manner (*Figure 5*). Last, *Dilp3* null mutations increased sleep amounts (*Figure 2D*). Based on these data, we propose a model in which the Hugin/PK2-R1/Dilps axis along the HugPC-IPC circuitry modulates larval sleep (*Figure 7*).

Examples of Hugin acting through PK2-R1 are prominent in the context of feeding regulation in larvae and adults (*Schoofs et al., 2014*). For example, a recent study has suggested that Hugin regulates the timing of pupariation and body wall contraction during pupariation through PK2-R1 in PTTH neurons (*Ohhara et al., 2024*). In the present study, we showed that HugPC in the larval brain modulates sleep (*Figure 3F*). A previous electron microscopic analysis suggested that larval HugPC neurons likely form synaptic connections to downstream neurons including IPCs (*Hückesfeld et al.,*

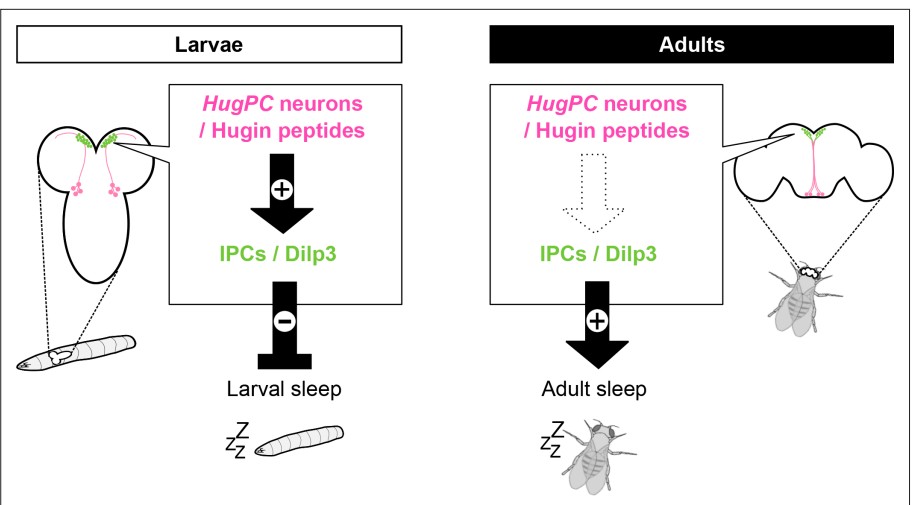

**Figure 7.** A schematic model of the Hugin/PK2-R1/Dilps axis in larval and adult sleep. Schematic summary comparing larvae and adults. In larvae, bath application of Hugin peptides activates insulin-producing cells (IPCs), and both Hugin signaling (HugPC neurons/Hugin peptides) and IPC output (including Dilp3) act to suppress larval sleep. In adults, Hugin peptides do not activate IPCs (dashed arrow), whereas IPC/Dilp3 signaling promotes adult sleep.

*2021*; *Schlegel et al., 2016*). We here showed that PK2-R1 is required for larval sleep control and that thermogenetic activation of Hug neurons is associated with Ca$^{2+}$ responses in PK2-R1-expressing IPCs in vivo (*Figure 4*). Importantly, our ex vivo data indicated that Hugin peptide-induced Ca$^{2+}$ response in IPCs was largely abolished by *PK2-R1* mutation (*Figure 5*), indicating the critical role of the Hugin/ PK2-R1 axis for the HugPC-mediated IPC activation. Together, we propose that the HugPC-IPCs circuity mobilizes Hugin/PK2-R1 signaling to selectively modulate larval sleep. Interestingly, our ex vivo data indicated that NMU, a mammalian ortholog of Hugin, can trigger Ca$^{2+}$ responses in larval IPCs in a PK2-R1-dependent fashion (*Figure 5*). Given that NMU has been implicated in sleep regulation in fishes and mammals (*Ahnaou and Drinkenburg, 2011*; *Chiu et al., 2016*), an analogous ligand/ receptor pair might be involved in sleep modulation in mammals as well.

In this study, activation of Hugin–PK2-R1 signaling by bath application of Hugin peptides led to a significant reduction in Dilp3 immunoreactivity in IPCs, whereas Dilp2 immunoreactivity did not show a decrease under these conditions, suggesting that Hugin–PK2-R1 activation promotes Dilp3 secretion without strongly enhancing Dilp2 release in our assay. One plausible explanation is that individual Dilps are stored in partially distinct vesicular pools within IPCs and are released through stimulus-specific mechanisms. Consistent with this idea, previous work has shown that Dilp release from IPCs can be selective, with Dilp2 and Dilp3 responding to distinct nutritional cues (*Kim and Neufeld, 2015*). Confocal analyses in that study further demonstrated that Dilp2 and Dilp3 segregate into different intracellular granules, supporting the notion that individual Dilps can be targeted to distinct secretory pathways within the same neurons. In light of these observations, the selective effect of Hugin–PK2-R1 activation on Dilp3, but not Dilp2, is readily interpreted as another example of non-uniform Dilp regulation. Rather than uniformly driving secretion of all IPC-derived Dilps, we propose that Hugin–PK2-R1 signaling preferentially mobilizes the Dilp3 secretory pathway. Such selective regulation is reminiscent of piecemeal degranulation in mammalian eosinophils and mast cells, as well as stimulus-dependent mobilization of specific vesicle pools in pancreatic β cells. In this model, different Dilps are stored in partially distinct vesicular pools and are mobilized by different upstream inputs. While ultrastructural approaches such as electron microscopy could provide further insight into the anatomical segregation of Dilp-containing vesicles, these analyses are technically beyond the scope of the present study, and we therefore highlight them as an important direction for future work.

PK2-R1 knockout larvae exhibited both an increase in sleep and a reduction in locomotion speed. Because our locomotor metric is defined as travel distance per unit time during epochs classified as wake, the observed decrease in locomotion speed cannot be fully explained by the increased fraction of time spent in sleep. Nevertheless, we cannot exclude the possibility that part of the apparent sleep

increase in the knockout reflects broader motor impairments in PK2-R1-expressing circuits. PK2-R1 expression is not confined to IPCs but is broadly distributed in the nervous system, including populations in the ventral nerve cord (*Figure 1C*), making it likely that PK2-R1-positive neurons outside IPCs contribute to locomotor control. Consistent with this idea, when we restricted our manipulations to IPCs, we observed robust changes in sleep without significant alterations in locomotor activity (*Figure 2—figure supplement 1*). These findings suggest that IPCs are more directly involved in sleep regulation, whereas other PK2-R1-expressing neurons may predominantly influence locomotion.

Dh44 neurons promote arousal in second-instar larvae (*Poe et al., 2023*), raising the possibility that Hugin signaling may influence sleep through Dh44 in addition to IPCs. Consistent with this idea, the Hugin receptor PK2-R1 is expressed not only in IPCs but also in Dh44 and DMS neurons (*Schlegel et al., 2016*). Thus, an extension of our model is that HuginPC-derived Hugin may act on multiple PK2-R1-positive targets (including IPCs and potentially Dh44/DMS neurons) to shape larval sleep. Future cell type-specific perturbations of PK2-R1 in Dh44 and DMS neurons will be important to test the contribution of these pathways.

## Divergent roles of insulin signaling in regulation of larval sleep and feeding

IPCs in larvae respond to exogenously applied Hug peptides, both by exhibiting elevated $Ca^{2+}$ levels and decreases in intracellular Dilp3 storage (*Figure 5C–E*). In contrast to Dilp3, however, the amount of Dilp2 accumulated in IPCs remains unchanged following Hug peptide application (*Figure 5—figure supplement 1B*). On the other hand, Dilp2 accumulation in IPCs decreases after glucose treatment (*Figure 5—figure supplement 1A*; *Oh et al., 2019*). These results unexpectedly suggest that larval IPCs form parallel channels inside, one for Hug-Dilp3 and the other for glucose. Mechanistically, Hug acts directly on IPCs via the PK2-R1 receptor. As for glucose, on the other hand, glucose-sensing CN neurons in the brain take up glucose and subsequently secrete another neuropeptide to activate IPCs (*Oh et al., 2019*). Such parallelism inside IPCs might provide insight into why insect species broadly express multiple insulin-like peptide genes, in some cases even as many as 40 genes in the silkworm Bombyx mori (*Okamoto, 2021*). *Drosophila* genome harbors eight Dilp genes (*Nässel et al., 2015*), of which Dilp2, Dilp3, and Dilp5 are considered nutrient-related. Nevertheless, mRNA expression or release patterns of these Dilps are not identical under different nutrient conditions (*Géminard et al., 2009*; *Kauffman and DiAngelo, 2024*; *Nässel et al., 2015*; *Oh et al., 2019*; *Ikeya et al., 2002*). Furthermore, these Dilps exhibit different binding affinities for the insulin receptor (InR), the secreted decoy of InR (SDR), the ecdysone-inducible gene L2 (Imp-L2) (*Okamoto et al., 2013*). These reports hint at the differential roles of different insulin-like peptides, but mechanistic insights remain largely lacking. Our data of larval IPCs forming parallel channels for Hug-Dilp3 and glucose-Dilp2 may provide a point of entry for tackling this mystery.

## Differential impacts of the same neuropeptidergic modulations on larval and adult sleep

Our data show that the Hugin/PK2-R1/Dilps axis likely downregulates sleep amounts in larvae. Despite the significant increase of sleep amount in *Hugin* mutant larvae, *Hugin* mutant adults showed sleep amounts comparable to wild-type control (*Figure 6A*), consistent with a previous report that Hugin is dispensable for control of baseline sleep levels (*Schwarz et al., 2021*). According to our histological analyses, the Hugin/PK2-R1 expression patterns as well as the HugPC-IPCs circuitry appear to be largely conserved in adults (*Figure 3E*, *Figure 6—figure supplement 2*). Unlike the larval brain, however, Hugin peptides failed to trigger $Ca^{2+}$ responses in IPCs in adults (*Figure 6E*). It is thus possible that Hugin/PK2-R1 signaling along the HugPC-IPCs circuitry is suppressed in adults. IPCs in adults receive multiple positive and negative modulatory inputs through GPCRs including the metabotropic $GABA_B$ receptors (*Enell et al., 2010*), which suppress IPC activity and Dilp release in adult IPCs (*Enell et al., 2010*). It is thus plausible that such negative modulatory inputs to IPCs in adults might counteract with the Hugin/PK2-R1 axis to suppress Dilp release. In addition, our data suggest that Dilps modulate sleep amount in the opposite directions in larvae and adults (*Figure 7*). Comparing the expression levels and activities of GPCRs in larval and adult IPCs would be essential to better understand how the same modulatory signals over the course of development come to exert differential impacts on sleep. Interestingly, Hugin in adults appears irrelevant for the baseline sleep

amount but is required for homeostatic regulation of sleep (*Schwarz et al., 2021*). Thus, testing if the Hugin/PK2-R1 axis is involved in the homeostatic regulation of larval sleep, and how such a system compares to its adult counterpart, may further provide mechanistic insights into how homeostatic sleep regulation matures over development.

In summary, we identified the Hugin/PK2-R1/Dilps signaling along the HugPC/IPCs circuit as the molecular and circuitry basis for developmental sleep regulation. Our study begins to fill the knowledge gap in larval sleep regulation and sheds light on mechanistic differences in sleep regulation between early developmental stages and adults.

# Materials and methods

**Key resources table**

| Reagent type (species) or resource | Designation | Source or reference | Identifiers | Additional information |
|---|---|---|---|---|
| Gene (*Drosophila melanogaster*) | PK2-R1 | FlyBase | FLYB:FBgn0038140 | |
| Gene (*D. melanogaster*) | Dilp3 | FlyBase | FLYB:FBgn0044050 | |
| Gene (*D. melanogaster*) | Dilp5 | FlyBase | FLYB:FBgn0044048 | |
| Gene (*D. melanogaster*) | Hug | FlyBase | FLYB:FBgn0028374 | |
| Gene (*Homo sapiens*) | NMU | HGNC | HGNC:7859 | |
| Genetic reagent (*D. melanogaster*) | w[1118] | Other | NA | Provided by Dr Yi Rao, Peking University |
| Genetic reagent (*D. melanogaster*) | iso31 | Bloomington *Drosophila* Stock Center | BDSC:5905; RRID:BDSC_5905 | Isogenized for chr 1;2;3, with w[1118] line |
| Genetic reagent (*D. melanogaster*) | PK2-R1[attP] | Bloomington *Drosophila* Stock Center | BDSC:84563; RRID:BDSC_84563 | FlyBase symbol: TI{TI}PK2-R1[attP] |
| Genetic reagent (*D. melanogaster*) | PK2-R1[2A-GAL4] | Bloomington *Drosophila* Stock Center | BDSC:84686; RRID:BDSC_84686 | FlyBase symbol: TI{2A-GAL4}PK2-R1[2A-GAL4] |
| Genetic reagent (*D. melanogaster*) | PK2-R1[2A-LexA] | Bloomington *Drosophila* Stock Center | BDSC:84431; RRID:BDSC_84431 | FlyBase symbol: TI{2A-lexA::GAD}PK2-R1[2A-lexA] |
| Genetic reagent (*D. melanogaster*) | Dilp3[1] | Bloomington *Drosophila* Stock Center | BDSC:30882; RRID:BDSC_30882 | FlyBase symbol: TI{TI}Ilp3[1] |
| Genetic reagent (*D. melanogaster*) | Dilp5[1] | Bloomington *Drosophila* Stock Center | BDSC:30884; RRID:BDSC_30884 | FlyBase symbol: TI{TI}Ilp5[1] |
| Genetic reagent (*D. melanogaster*) | Dilp3-GAL4 | Bloomington *Drosophila* Stock Center | BDSC:52660; RRID:BDSC_52660 | FlyBase symbol: P{Ilp3-GAL4.C}2 |
| Genetic reagent (*D. melanogaster*) | Dilp5-GAL4 | Bloomington *Drosophila* Stock Center | BDSC:66007; RRID:BDSC_66007 | FlyBase symbol: P{Ilp5-GAL4.L}8 |
| Genetic reagent (*D. melanogaster*) | Hug[attP] | Bloomington *Drosophila* Stock Center | BDSC:84514; RRID:BDSC_84514 | FlyBase symbol: TI{TI}Hug[attP] |
| Genetic reagent (*D. melanogaster*) | Hug[2A-GAL4] | Bloomington *Drosophila* Stock Center | BDSC:84646; RRID:BDSC_84646 | FlyBase symbol: TI{2A-GAL4}Hug[2A-GAL4] |
| Genetic reagent (*D. melanogaster*) | Hug[2A-LexA] | Bloomington *Drosophila* Stock Center | BDSC:84397; RRID:BDSC_84397 | FlyBase symbol: TI{2A-lexA::GAD}Hug[2A-lexA] |
| Genetic reagent (*D. melanogaster*) | HugPC-GAL4 | *Hückesfeld et al., 2016* (10.1038/ncomms12796) | NA | Provided by Dr Michael J. Pankratz, University of Bonn |

*Continued on next page*

*Continued*

| Reagent type (species) or resource | Designation | Source or reference | Identifiers | Additional information |
|---|---|---|---|---|
| Genetic reagent (*D. melanogaster*) | HugPC-LexA | This paper | NA | A transgenic LexA driver generated in this study. A 544-bp Hug enhancer fragment was placed upstream of nlsLexA::p65 and inserted into the attP2 site by ΦC31 integrase-mediated transgenesis. See Materials and methods for details. |
| Genetic reagent (*D. melanogaster*) | UAS-Kir2.1::EGFP | Bloomington *Drosophila* Stock Center | BDSC:6596; RRID:BDSC_6596 | FlyBase symbol: P{UAS-Hsap\KCNJ2.EGFP}1 |
| Genetic reagent (*D. melanogaster*) | UAS-Stinger, LexAop-tdTomato.nls | Bloomington *Drosophila* Stock Center | BDSC:66680; RRID:BDSC_66680 | FlyBase symbol: P{UAS-Stinger}2; PBac{13XLexAop2-IVS-tdTomato.nls}VK00022 |
| Genetic reagent (*D. melanogaster*) | UAS-mCD8::mCherry, UAS-CRTC::GFP | Bloomington *Drosophila* Stock Center | BDSC:99657; RRID:BDSC_99657 | FlyBase symbol: P{UAS-mCD8.mCherry-T2A-lacZ.nls}JK22C; P{UAS-Crtc.GFP}attP40 |
| Genetic reagent (*D. melanogaster*) | UAS-ReaChr | Bloomington *Drosophila* Stock Center | BDSC:53741; RRID:BDSC_53741 | FlyBase symbol: P{UAS-ReaChR}attP40 |
| Genetic reagent (*D. melanogaster*) | UAS-TrpA1 | Bloomington *Drosophila* Stock Center | BDSC:26263; RRID:BDSC_26263 | FlyBase symbol: P{UAS-TrpA1(B).K}attP16 |
| Genetic reagent (*D. melanogaster*) | LexAop-TrpA1 | *Burke et al., 2012* (10.1038/nature11614) | NA | Provided by Dr Scott Waddel, University of Oxford |
| Genetic reagent (*D. melanogaster*) | LexAop-rCD2::RFP-p10. UAS-mCD8::GFP-p10 | Bloomington *Drosophila* Stock Center | BDSC:67093; RRID:BDSC_67093 | FlyBase symbol: P{lexAop-rCD2::RFP-p10.UAS-mCD8::GFP-p10}su(Hw)attP5 |
| Genetic reagent (*Escherichia coli*) | One Shot ccdB Survival 2 T1ᴿ competent cells | Thermo Fisher Scientific | Thermo Fisher Scientific:A10460 | |
| Antibody | Anti-GFP (chicken polyclonal) | Aves Labs | Aves Labs:GFP-1020; RRID:AB_10000240 | IHC (1:1000) |
| Antibody | Anti-mCherry (mouse monoclonal) | Takara Bio | Takara Bio:632543; RRID:AB_2307319 | IHC (1:1000) |
| Antibody | Anti-Dilp3 (rabbit polyclonal) | *Veenstra et al., 2008* (10.1007/s00441-009-0769-y) | NA | IHC (1:250) Provided by Dr Jan A. Veenstra, Université de Bordeaux |
| Antibody | Anti-Dilp2 (rabbit polyclonal) | *Okamoto et al., 2012* (10.1073/pnas.1116050109) | NA | IHC (1:2000) Provided by Dr Takashi Nishimura, RIKEN |
| Antibody | Anti-brp (mouse monoclonal) | Developmental Studies Hybridoma Bank | DSHB:nc82; RRID:AB_2314866 | IHC (1:100) |
| Antibody | Anti-histone H3 phospho S28 (rat monoclonal) | Abcam | Abcam:ab10543; RRID:AB_2295065 | IHC (1:1000) |
| Antibody | Goat anti-chicken Alexa Fluor 488 | Thermo Fisher Scientific | Thermo Fisher Scientific:A-11039; RRID:AB_2534096 | IHC (1:200) |
| Antibody | Goat anti-mouse Alexa Fluor 555 | Thermo Fisher Scientific | Thermo Fisher Scientific:A-21424; RRID:AB_141780 | IHC (1:200) |
| Antibody | Goat anti-rabbit Alexa Fluor 633 | Thermo Fisher Scientific | Thermo Fisher Scientific:A-21071; RRID:AB_2535732 | IHC (1:200) |
| Antibody | Goat anti-rat Alexa Fluor 633 | Thermo Fisher Scientific | Thermo Fisher Scientific:A-21094; RRID:AB_2535749 | IHC (1:200) |

*Continued*

| Reagent type (species) or resource | Designation | Source or reference | Identifiers | Additional information |
|---|---|---|---|---|
| Recombinant DNA reagent | pBPnlsLexA::p65Uw | Addgene | Addgene:26230; RRID:Addgene_26230 | |
| Sequence-based reagent | HugPC enhancer | This paper | NA | A 544-bp Hug enhancer fragment used in this study for transgene construction. PCR-amplified from *Drosophila* genomic DNA. See Materials and methods for details. |
| Peptide, recombinant protein | Hug-γ | This paper | NA | Chemically synthesized *Drosophila* Hug-γ peptide with amino acid sequences of Acetyl-QLQSNGEPAYRVRTPRL-CONH2. See Materials and methods for details. |
| Peptide, recombinant protein | PK-2 | This paper | NA | Chemically synthesized *Drosophila* PK-2 peptide with amino acid sequences of Acetyl-SVPFKPRL-CONH2. See Materials and methods for details. |
| Peptide, recombinant protein | NMU | This paper | NA | Chemically synthesized human NMU peptide with amino acid sequences of Acetyl-FRVDEEFQSPFASQSRGYFL FRPRN-CONH2. See Materials and methods for details. |
| Commercial assay or kit | PrimeSTAR Max DNA Polymerase | Takara Bio | Takara Bio:R045A | |
| Commercial assay or kit | In-Fusion HD Cloning Kit w/Cloning Enhancer | Takara Bio | Takara Bio:639635 | |
| Chemical compound, drug | D(+)-Glucose | FUJIFILM Wako Pure Chemical Corporation | FUJIFILM Wako Pure Chemical Corporation:041-00595 | |
| Software, algorithm | Adobe Illustrator 2023 | Adobe | RRID:SCR_010279 | |
| Software, algorithm | Adobe Photoshop 2023 | Adobe | RRID:SCR_014199 | |
| Software, algorithm | ImageJ v1.53t | *Schneider et al., 2012* (10.1038/nmeth.2089) | RRID:SCR_003070 | |
| Software, algorithm | Inkscape v1.2 | The Inkscape Team | RRID:SCR_014479 | |
| Software, algorithm | Prism v9.5.1 | GraphPad | RRID:SCR_002798 | |
| Software, algorithm | Python Programming Language v3.10.0 | Python Software Foundation | RRID:SCR_008394 | |
| Software, algorithm | R Project for Statistical Computing v4.1.2 | R Core Team 2021 | RRID:SCR_001905 | |
| Software, algorithm | SnapGene v5.3.3 | GSL Biotech LLC | RRID:SCR_015052 | |
| Software, algorithm | Arduino UNO | Arduino | RRID:SCR_017284 | |
| Software, algorithm | TriKinetics DAMSystem3 Software | TriKinetics Inc (available at https://trikinetics.com/) | RRID:SCR_021809 | |
| Other | TriKinetics *Drosophila* Activity Monitoring System | TriKinetics Inc | RRID:SCR_021798 | |

| Reagent type (species) or resource | Designation | Source or reference | Identifiers | Additional information |
|---|---|---|---|---|
| Other | Fan-less Peltier-type incubator | Mitsubishi Electric Engineering Co, Ltd | Mitsubishi Electric Engineering Co, Ltd: SLC-25 | Used to keep flies for DAM experiments |

## Fly strains

The following strains of *Drosophila melanogaster* were obtained from Bloomington *Drosophila* Stock Center: *iso31* (an isogenic *w*[1118] strain; BDSC# 5905), *PK2-R1*[attP] (BDSC# 84563), *PK2-R1*[2A-GAL4] (BDSC# 84686), *PK2-R1*[2A-LexA] (BDSC# 84431), *Dilp3*[1] (BDSC# 30882), *Dilp5*[1] (BDSC# 30884), *Dilp3-GAL4* (BDSC# 52660), *Dilp5GAL4* (BDSC# 66007), *Hug*[attP] (BDSC# 84514), *Hug*[2A-GAL4] (BDSC# 84646), *Hug*[2A-LexA] (BDSC# 84397), *UAS-Kir2.1::EGFP* (BDSC# 6596), *UAS-Stinger, LexAoptdTomato.nls* (BDSC# 66680), *UAS-mCD8::mCherry, UAS-CRTC::GFP* (BDSC# 99657), *UAS-ReaChR* (BDSC# 53741), *UAS-TrpA1* (BDSC# 26263), *LexAop-ReaChR* (BDSC# 53746), *LexAop-rCD2::RFP-p10.UAS-mCD8::GFP-p10* (BDSC# 67093). *w*[1118] was from Y. Rao (Peking University); *HugPC-GAL4* (*Hückesfeld et al., 2016*) was from M. Pankratz (University of Bonn); *LexAop-TrpA1* (*Burke et al., 2012*) was from S. Waddel (University of Oxford). *HugPC-LexA* was created in this study (see Generation of transgenic lines). Detailed lists of fly genotypes can be found in the key resource table and Source Data files.

## General fly maintenance

Flies were maintained in an incubator (PHCbi, model# MTR-554-PJ) set to 25°C with a humidity of over 60% and a 9 AM:9 PM light/dark cycle. After 10 d of rearing in bottles, old flies were replaced with freshly eclosed ones to maintain the strain (*Furusawa et al., 2023*).

## Egg collection

Agar substrates for egg collection were prepared by microwave-heating a mixture of 2.0 g sucrose (Fujifilm Wako Pure Chemicals, cat# 196-00015), 3.0 g agar (BD, cat# 214010), 2.5 ml apple juice (Dole), and Milli-Q water up to 100 ml. Melted agar was poured into Petri dishes with a diameter of 4 cm (approximately 3 ml per dish). Solidified agar was scratched with tweezers to make the surface rough. Agar plates were stored inside a sealed container at 4°C. Upon egg collection, yeast paste (dried yeast (Nippon Beet Sugar Co, Ltd) kneaded with Milli-Q water at 40% (wt/wt)) was placed in the center of the agar plates to further promote oviposition.

## Larval staging

First-instar larvae were collected 2 days after egg collection. Among them, second-instar larvae that molted within a 2-hr window were used for video recording. Distinguishing between first- and second-instar larvae was based on morphological characteristics: while anterior spiracles were not evident in the first instar, fist-shaped spiracles were observed in the second instar (*Lakhotia and Ranganath, 2021*).

## Larval sleep analysis

Twenty-four-well silicone chambers for sleep analysis, described in *Figure 1—figure supplement 1*, were made by pouring dimethylpolysiloxane into molds based on a previous study (*Churgin et al., 2019*; *Szuperak et al., 2018*). As a substrate, 120 µl of a mixture of 3% agar and 2% sucrose was poured into each well. After the gel solidified, 10 µl per well of 60 mg/ml yeast suspension was applied to the surface of the gel. Larvae were placed individually into the wells, and a glass plate (Azone, cat# 14540-01, 5 mm × 200 mm × 200 mm) was positioned on top of the chamber to prevent larvae from escaping. The chamber was placed inside a dark box equipped with a ring-type infrared LED (CCS, cat# LDR2-132IR940-LA) and a camera (The Imaging Source, cat# DMK27BUP031), and recording was started immediately without acclimation. Images were captured through the IC Capture software (The Imaging Source, version 2.5) under the following conditions: MJPEG Compressor for codec, a frame rate of 0.87 fps, and an image size of Y800 (2592 × 1944). Locomotor activity was quantified from the same video recordings used for larval sleep analysis. Travel distance was calculated as the cumulative displacement of the larval centroid over time, and locomotion speed was defined as total

travel distance divided by the total duration of wakefulness, considering only time bins classified as wake. To minimize the possibility that feeding behavior was misclassified as sleep, we adjusted our immobility threshold using video recordings in which larval feeding bouts, including relatively subtle episodes, were apparent, and configured the algorithm so that feeding-associated movements would generally be classified as wake rather than inactivity. During the development and routine use of the assay, we also inspected representative recordings across genotypes and did not observe obvious feeding-related abnormalities or gross locomotor defects under our HuginPC/IPC manipulations.

## Sleep deprivation

Blue light irradiation was performed using a 455-nm LED (Thorlabs, cat# M455L4) with a collimating adapter (Thorlabs, cat# SM1U25-A) (*Omamiuda-Ishikawa et al., 2020*; *Yoshino et al., 2017*; *Yoshino et al., 2025*). The irradiation intensity was set to the maximum scale using a T-cube LED driver (Thorlabs, cat# LEDD1B). Temporal control of LED illumination was achieved using the Arduino UNO microcontroller board (Arduino, RRID:SCR_017284). For sleep deprivation experiments, the first 1 hr was recorded without LED illumination, followed by another 1 hr consisting of repeated cycles of 'LED ON for 90 s and OFF for 30 s'.

## Immunohistochemistry

Antibody staining of larval neurons was carried out as previously reported (*Morikawa et al., 2011*; *Omamiuda-Ishikawa et al., 2020*). Briefly, dissection in phosphate-buffered saline (PBS); fixation in 4% paraformaldehyde/PBS at room temperature for either 90 min in CRTC::GFP experiments or 30 min in the others; washing with 0.3% Triton X-100/PBS (hereafter referred to as PBST) for 10 min × 3; blocking in 5% normal goat serum (NGS)/PBST at room temperature for 30 min; primary antibody treatment at 4°C overnight; washing with PBST for 10 min × 3; secondary antibody treatment at 4°C overnight; washing with PBST for 10 min × 3; and mounting in VECTASHIELD medium (Vector Laboratories, cat# H-1000, RRID:AB_2336789). Reactions were performed either in Terasaki 96-well plates (Stem Corp., cat# P96R37N) for adults or in 48-well culture plates (Corning, cat# 3548) for larvae. The following primary antibodies were diluted in 5% NGS/PBST: chicken polyclonal anti-GFP (1:1000, Aves Labs, cat# GFP-1020, RRID:AB_10000240), mouse monoclonal anti-mCherry (1:1000, Takara Bio, cat# 632543, RRID:AB_2307319), rabbit polyclonal anti-Dilp3 (1:250, *Veenstra et al., 2008*), rabbit polyclonal anti-Dilp2 (1:2000, *Okamoto et al., 2012*), mouse monoclonal anti-BRP (1:100, Developmental Studies Hybridoma Bank, clone nc82; RRID:AB_2314866), and rat monoclonal anti-histone H3 phospho S28 (1:1000, Abcam, cat# ab10543; RRID:AB_2295065). The following secondary antibodies were diluted in 5% NGS/PBST: goat anti-chicken Alexa Fluor 488 (1:200, Thermo Fisher Scientific, cat# A-11039, RRID:AB_2534096), goat anti-mouse Alexa Fluor 555 (1:200, Thermo Fisher Scientific, cat# A-21424, RRID:AB_141780), goat anti-rabbit Alexa Fluor 633 (1:200, Thermo Fisher Scientific, cat# A-21071, RRID:AB_2535732), and goat anti-rat Alexa Fluor 633 (1:200, Thermo Fisher Scientific, cat# A-21094; RRID:AB_2535749). Fluorescence images were obtained using a confocal microscope (Leica, TCS SP8) with 20× (Leica, HC PL APO CS2 20×/0.75 IMM) and 63× (Leica, HC PL APO CS2 63×/1.40 OIL) objectives. Images were captured at 512 × 512-pixel resolution with an interval of 1 μm. Z-stack images of the maximum projections were generated using the ImageJ v1.53t software (*Schneider et al., 2012*; RRID:SCR_003070).

## Generation of transgenic lines

*HugPC-LexA* lines were generated by ΦC31 integrase-mediated transgenesis, as previously described (*Groth et al., 2004*).

The 544-bp enhancer region of *Hug*, essentially the same as that used to generate *HugPC-GAL4* (*Hückesfeld et al., 2016*), was PCR-amplified from fly genomic DNA. The gel-purified amplicon was subjected to another round of PCR to add hangout sequences. The backbone suicide vector pBPnlsLexA::p65Uw (RRID:Addgene_26230) was initially amplified by transforming the *ccdB*-resistant *E. coli* strain, One Shot *ccdB* Survival 2 T1[R] competent cells (Thermo Fisher Scientific, cat# A10460). The PCR product and the vector were double-digested by Aat II (NEB, cat# R0117S) and Fse I (NEB, cat# R0588S), followed by ligation using In-Fusion HD Cloning Kit w/Cloning Enhancer (Takara Bio, cat# 639635). The integrity of the resulting plasmid was confirmed by DNA sequencing. Plasmid insertion was targeted to four attP sites (attP2, attP40, VK00005, and su(Hw)attP5) individually through ΦC31

integrase-mediated transgenesis by BestGene Inc (Chino Hills, CA). Transformants were selected based on the eye color marker. Among the generated strains harboring *HugPC-LexA* in each insertion site, labeling patterns in the larval brain were checked by driving fluorescent marker expression under the control of *LexAop* (**Figure 3—figure supplement 2**). As the *HugPC-LexA* inserted in attP2 closely resembled the labeling pattern of the original *HugPC-GAL4*, it was used in further experiments.

The following primers were used for cloning:

> Forward primer for the *HugPC* enhancer; first round PCR against genomic DNA: 5′-AAGGGTTT GGTTTAATTTATTTATGTCATA-3′
>
> Reverse primer for the *HugPC* enhancer; first round PCR against the genomic DNA: 5′-GGAC AACTGATGCCAGCAGC-3′
>
> Forward primer for the *HugPC* enhancer with hangout sequences (underlined); second round PCR against the first round amplicon: 5′-gaaaagtgccacctgacgtAAGGGTTTGGTTTAATTTATTTAT GTCATA-3′
>
> Reverse primer for the *HugPC* enhancer with hangout sequences (underlined); second round PCR against the first round amplicon: 5′-cccgggcgagctcggccggGGACAACTGATGCCAGCAGC -3′

## Food intake assay

Food intake was measured using a dye-based feeding assay adapted from *Hückesfeld et al., 2016*. Apple juice agar plates were prepared with a spot of red yeast paste in the center. After 30 min starvation, larvae were transferred onto the yeast paste and videotaped for 20 min. Larvae were then collected in a small cell strainer and rinsed with 60°C hot water to remove external dye, transferred to glass slides, and photographed. Images were analyzed in ImageJ by measuring the area of red-stained dye signal and the total body area, and food intake was quantified as the ratio of dye-stained area to total body area for each larva.

## Measurement of neuronal activity and Dilp3 accumulation in IPCs

The GFP-tagged CRTC (CRTC::GFP) reporter was applied to quantify the activities of larval IPCs, essentially according to the previous report in adult flies (*Bonheur et al., 2023*).

Parental *Dilp3-GAL4* and the reporter line harboring *UAS-CRTC::GFP* and *UASmCD8::mCherry* (BDSC# 99657) were crossed, and eggs were collected within an 8- to 10-hr time window. Offspring larvae were reared for 4 days after egg laying (AEL) at 25°C to early third instar. Collected larvae were washed with water and starved on water-soaked paper for 12–14 hr. For in vivo thermogenetic activation, larva-containing Petri dishes were warmed at 30°C for 60 min. For ex vivo thermogenetic experiments, larvae were dissected in calcium- and sugar-free HL3.1 buffer (*Nakamizo-Dojo et al., 2023*) at 30°C, and each brain was incubated in the same buffer at 30°C for 5 min. For ex vivo glucose or peptide application assays, larvae were first dissected in the same buffer as above at room temperature, and brain samples were then treated with either 20 mM D-glucose or 1 mM of chemically synthesized peptides dissolved in HL3.1 buffer for 5 min. After incubation, samples were subjected to immunohistochemical analyses to visualize GFP, mCherry, and Dilp3 signals (see Immunohistochemistry). Image processing was performed using ImageJ v1.53t (*Schneider et al., 2012*; RRID:SCR_003070). Nuclear ROIs and cell body outlines were drawn based on the mCD8::mCherry fluorescence insensitive to neuronal activity, and the mean intensities of CRTC::GFP signals within the nucleus ($F_{nuc}$) and the cytoplasm ($F_{cyto} = F_{whole\ cell} - F_{nuc}$) were quantified. NLI was calculated using the following formula: ($F_{nuc} - F_{cyto}$)/($F_{nuc} + F_{cyto}$). To assess Dilp3 accumulation in IPCs, the anti-Dilp3 signal intensity was measured within the cytosol of each cell, excluding the nucleus. Data from 4 to 14 Dilp3-positive IPCs per brain, collected from 3 to 7 larvae, were pooled for each genotype/treatment group.

We attempted to record GCaMP-based calcium dynamics in Hugin neurons and IPCs, but did not pursue this approach further because it was low-throughput and a positive control assay (glucose application) did not elicit the expected IPC calcium response in our hands.

## Chemical synthesis of Hug peptides

Peptides with the following amino acid sequences were chemically synthesized using the Fmoc solid phase method, and their purity was assessed by HPLC to be over 95%.

All procedures were carried out by Eurofins Genomics K.K. (Tokyo, Japan).

> *Drosophila* Hug-γ: Acetyl-QLQSNGEPAYRVRTPRL-CONH2 (17 a.a.)
> *Drosophila* PK-2: Acetyl-SVPFKPRL-CONH2 (8 a.a.)
> Human NMU: Acetyl-FRVDEEFQSPFASQSRGYFLFRPRN-CONH2 (25 a.a.)

## Sleep quantification in adult flies

The following parts of the *Drosophila* Activity Monitor (DAM) system were purchased from TriKinetics Inc (Waltham, MA): the DAM (cat# DAM2), the power supply interface unit (cat# PSIU9), monitor tubes with an outside diameter of 5 mm and a length of 65 mm (cat# PGT5x65), and tube caps (cat# CAP5-Black). Newly eclosed adult flies were collected under brief $CO_2$ anesthesia and kept as groups of 10–15 in standard food vials. Males and females were kept separately to avoid mating. To minimize disturbance by noise and vibration, flies were placed inside a fan-less Peltiertype incubator (Mitsubishi Electric Engineering Co, Ltd, cat# SLC-25) set to 25°C, externally connected with 5 mm diameter LED lights (MY-CRAFT, Ltd., cat# 5W0601) and a timer device (Panasonic Corp, cat# WH3311WP) to generate a 9 AM:9 PM light/dark cycle. After 3 days of eclosion, each fly was gently aspirated into a Pyrex glass monitor tube, one end of which was stuffed with agar food (2% (wt/wt) agar, 5% (wt/wt) sucrose), and the other was plugged with a cotton string. The tubes were set to the monitor device and placed inside the same fan-less incubator. The DAM system was connected to an external PC installed with DAMSystem3 Data Collection Software (RRID:SCR_021809; available at https://trikinetics.com/) for data acquisition, and beam-crossing by each fly was counted for two successive days. Time windows longer than 5 min without beam-crossing were judged as sleep bouts, following the conventional criteria for adult fly sleep (*Shaw et al., 2000*).

## Statistical analysis

Sample sizes were not predetermined by formal power analysis and were based on prior studies and experimental feasibility. Statistical analyses by Kruskal–Wallis one-way ANOVA, Mann–Whitney *U*-test, or chi-square test were performed using Prism 9.5.1 (GraphPad, RRID:SCR_002798). Bonferroni correction was applied for multiple comparisons. No formal randomization procedure was applied. Analyses were performed blinded to genotype or experimental condition for all datasets except the CRTC imaging data, for which the condition was known during analysis due to procedural constraints. Asterisks (*) represent p-values as indicated within each figure legend. All data necessary to reproduce the figure panels and statistical analyses are available as Source Data files.

## Acknowledgements

We thank Y Rao, M Pankratz, S Waddel, and the Bloomington Drosophila Stock Center for fly stocks; JA Veenstra for the anti-Dilp3 antibody; T Nishimura for the anti-Dilp2 antibody; Y Nishizuka for valuable suggestions on writing code for automated larval detection and sleep quantification; H Ito, N Yamaji, M Maetani, A Ogasawara, E Kato, T Rikiishi, S Ando, M Hayashi, and S Miyazaki for technical assistance and fly maintenance; and the members of Emoto Lab for critical comments and discussion. This work is supported by MEXT Grants-in-Aid for Scientific Research on Innovative Areas 'Dynamic regulation of brain function by Scrap and Build system' (KAKENHI 16H06456), JSPS (KAKENHI 16H02504), WPI-IRCN, AMED-CREST (JP22gm310010), JSTCREST, Toray Foundation, Naito Foundation, Takeda Science Foundation, and Uehara Memorial Foundation to KE; the Leading Initiative for Excellent Young Researchers (LEADER) from MEXT, JSPS (KAKENHI 22K06309), and AMED-PRIME (JP22gm6510011) to KI. Grant-in-Aid for JSPS Fellows (24KJ0911) to MM; and Grant-in-Aid for JSPS Fellows (22J21096; 22KJ1042) to CH.

# Additional information

## Funding

| Funder | Grant reference number | Author |
|---|---|---|
| Japan Society for the Promotion of Science | 22KJ1042 | Chikayo Hemmi |
| Japan Society for the Promotion of Science | 24KJ0911 | Mana Motoyoshi |
| Japan Society for the Promotion of Science | KAKENHI 22K06309 | Kenichi Ishii |
| Japan Agency for Medical Research and Development | JP22gm6510011 | Kenichi Ishii |
| Japan Society for the Promotion of Science | KAKENHI 16H06456 | Kazuo Emoto |
| Japan Society for the Promotion of Science | KAKENHI 16H02504 | Kazuo Emoto |
| Japan Agency for Medical Research and Development | JP22gm310010 | Kazuo Emoto |
| Japan Society for the Promotion of Science | 22J21096 | Chikayo Hemmi |

The funders had no role in study design, data collection, and interpretation, or the decision to submit the work for publication.

## Author contributions

Chikayo Hemmi, Conceptualization, Formal analysis, Investigation, Methodology, Resources, Validation, Visualization, Writing – original draft; Kenichi Ishii, Conceptualization, Data curation, Formal analysis, Funding acquisition, Investigation, Supervision, Validation; Mana Motoyoshi, Formal analysis, Writing – original draft, Resources; Masato Tsuji, Formal analysis, Software, Visualization; Kazuo Emoto, Data curation, Supervision, Funding acquisition, Conceptualization, Methodology, Writing – review and editing

## Author ORCIDs

Chikayo Hemmi ⓘ https://orcid.org/0009-0002-3815-8164
Kenichi Ishii ⓘ https://orcid.org/0000-0002-8834-5729
Kazuo Emoto ⓘ https://orcid.org/0000-0003-4194-801X

Reviewer #1 (Public review): https://doi.org/10.7554/eLife.105710.3.sa1
Reviewer #3 (Public review): https://doi.org/10.7554/eLife.105710.3.sa2
Author response https://doi.org/10.7554/eLife.105710.3.sa3

# Additional files

## Supplementary files

Supplementary file 1. Genotypes used in this study. Detailed genotypes for all fly lines used in the experiments.

MDAR checklist

Source data 1. Source data containing the raw data underlying all figures and statistical analyses in the manuscript.

## Data availability

All data necessary to reproduce the figures and statistical analyses are provided as *Source data 1* with the manuscript. The source code used in this study is available at https://doi.org/10.5281/zenodo.18996367.

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
