## [Editor Report · eLife Assessment]

The study investigates an emerging research field: the interaction between sleep and development. The authors used Drosophila larvae sleep as a study model and provide insight into how neuropeptide circuitry control sleep differentially between larvae and adult Drosophila. By using board range of behaviour and imaging methods and analysis, the authors provide a **valuable** investigation that demonstrates a larvae-specific sleep regulatory neural pathway of Hugin-PK2-Dilps in the Drosophila neurosecretory centre IPC. While some further text clarifications are still required, the revision presented **convincing** evidence supporting the claims with the new imaging data, sleep parametric analysis, and further clarification addressing the reviewers' comments.

---

## [Referee Report · Reviewer #1 (Public review)]

The manuscript investigates how neuropeptidergic signaling affects sleep regulation in Drosophila larvae. The authors first conduct a screen of CRISPR knock-out lines of genes encoding enzymes or receptors for neuropeptides and monoamines. As a result of this screen, the authors follow up on one hit, the hugin receptor, PK2-R1. They use genetic approaches including mutants and targeted manipulations of PK2-R1 activity in insulin-producing cells (IPCs) to increase total sleep amounts in 2nd instar larvae. Similarly, dilp3 and dilp5 null mutants and genetic silencing of IPCs show increases in sleep. The authors also show that hugin mutants and thermogenetic/optogenetic activation of hugin-expressing neurons caused reductions in sleep. Furthermore, they show through imaging-based approaches that hugin-expressing neurons activate IPCs. A key finding is that wash on of hugin peptides, Hug-γ and PK-2, in ex vivo brain preparations activates larval IPCs, as assayed by CRTC::GFP imaging. The authors then examine how the PK2-R1, hugin, and IPC manipulations affect adult sleep. Finally, the authors examine how Ca2+ responses through CRTC::GFP imaging in adult IPCs are influenced by the wash on of hugin peptides.

Strengths:

(1) This paper builds on previously published studies that examine Drosophila larval sleep regulation. Through the power of Drosophila genetics, this study yields additional insights into what role neuropeptides play in regulation of Drosophila larval sleep.

(2) This study utilizes several diverse approaches to examine larval and adult sleep regulation, neural activity, and circuit connections. The impressive array of distinct analyses provides new understanding into how Drosophila sleep-wake circuitry in regulated across the lifespan.

(3) The imaging approaches used to examine IPC activation upon hugin manipulation (either thermogenetic activation or wash on of peptides) demonstrate a powerful approach for examining how changes in neuropeptidergic signaling affect downstream neurons. These experiments involve precise manipulations as the authors use both in vivo and ex vivo conditions to observe an effect on IPC activity.

Weaknesses:

(1) There is limited discussion of why statistically significant differences are observed in some genetic and temperature controls. This discussion would better support the authors' conclusions.

(2) The functional connectivity of the huginPC-IPC circuit in larvae could be better supported by chemogenetics using real-time calcium imaging (GCaMP).

Comments on revisions:

I would like to thank the authors for the revisions. The inclusion of all sleep metrics, more detailed descriptions in the methods, & a more thorough comparison to other published articles has addressed most of my concerns.

---

## [Referee Report · Reviewer #3 (Public review)]

Summary:

Sleep affects cognition and metabolism, evolving throughout development. In mammals, infants have fast sleep-wake cycles that stabilize in adults via circadian regulation. In this study, the author performed a genetic screen for neurotransmitters/peptides regulating sleep and identified the neuropeptide Hugin and its receptor PK2-R1 as essential components for sleep in Drosophila larvae. They showed that IPCs express Pk2-R1 and silencing IPCs resulted in significant increase in the sleep amount, which was consistent with the effect they observed in PK2-R1 knock out mutants. They also showed that Hugin peptides, secreted by a subset of Hugin neurons (Hug-PC), activate IPCs through the PK2-R1 receptor. This activation prompts IPCs to release insulin-like peptides (Dilps), which are implicated in the modulation of sleep. They showed that Hugin peptides induce a PK2-R1 dependent calcium (Ca²⁺) increase in IPCs, which they linked to the release of Dilp3, showing a connection between Hugin signaling to IPCs, Dilp3 release and sleep regulation. Additionally, the activation of Hug-PC neurons reduced sleep amounts, while silencing them had the opposite effect. In contrast to the larval stage, the Hugin/PK2-R1 axis was not critical for sleep regulation in Drosophila adults, suggesting that this neuropeptidergic circuitry has divergent roles in sleep regulation across different stages of development.

Strengths:

This study used an updated system for sleep quantification in Drosophila larvae and this method allowed precise measurement of larval sleep patterns which is essential for the understanding of sleep regulation.

The authors performed unbiased genetics screening and successfully identified novel regulators for larval sleep, Hugin and its receptor PK2-R1, making a substantial contribution to the understanding of neuropeptidergic control of sleep regulation.

They clearly demonstrated the mechanism by which Hugin expressing neurons influence sleep through the activation of IPCs via PK2-R1 with Ca2+ responses and can modulate sleep.

Based on the demonstrated activation of PK2-R1 by the human Hugin orthologue Neuromedin U, research on human sleep disorders may benefit from the discoveries from Drosophila since sleep regulating mechanisms are conversed across species.

Weaknesses:

Previously identified weaknesses have been largely addressed by the authors.

---

## [Author Response]

The following is the authors’ response to the original reviews.

**Public Reviews:**

**Reviewer #1 (Public review):**
Summary:The study investigates how neuropeptidergic signaling affects sleep regulation in Drosophila larvae. The authors first conduct a screen of CRISPR knock-out lines of genes encoding enzymes or receptors for neuropeptides and monoamines. As a result of this screen, the authors follow up on one hit, the hugin receptor, PK2-R1. They use genetic approaches, including mutants and targeted manipulations of PK2-R1 activity in insulin-producing cells (IPCs) to increase total sleep amounts in 2nd instar larvae. Similarly, dilp3 and dilp5 null mutants and genetic silencing of IPCs show increases in sleep. The authors also show that hugin mutants and thermogenetic/optogenetic activation of hugin-expressing neurons caused reductions in sleep. Furthermore, they show through imaging-based approaches that hugin-expressing neurons activate IPCs. A key finding is that wash-on of hugin peptides, Hug-γ and PK-2, in ex vivo brain preparations activates larval IPCs, as assayed by CRTC::GFP imaging. The authors then examine how the PK2-R1, hugin, and IPC manipulations affect adult sleep. Finally, the authors examine how Ca2+ responses through CRTC::GFP imaging in adult IPCs are influenced by the wash-on of hugin peptides. The conclusions of this paper are somewhat well supported by data, but some aspects of the experimental approach and sleep analysis need to be clarified and extended.Strengths:(1) This paper builds on previously published studies that examine Drosophila larval sleep regulation. Through the power of Drosophila genetics, this study yields additional insights into what role neuropeptides play in the regulation of Drosophila larval sleep.(2) This study utilizes several diverse approaches to examine larval and adult sleep regulation, neural activity, and circuit connections. The impressive array of distinct analyses provides new understanding into how Drosophila sleep-wake circuitry in regulated across the lifespan.(3) The imaging approaches used to examine IPC activation upon hugin manipulation (either thermogenetic activation or wash-on of peptides) demonstrate a powerful approach for examining how changes in neuropeptidergic signaling affect downstream neurons. These experiments involve precise manipulations as the authors use both in vivo and ex vivo conditions to observe an effect on IPC activity.Weaknesses:Although the paper does have some strengths in principle, these strengths are not fully supported by the experimental approaches used by the authors. In particular:(1) The authors show total sleep amount over an 18-hour period for all the measures of 2nd instar larval sleep throughout the paper. However, published studies have shown that sleep changes over the course of 2nd instar development, so more precise time windows are necessary for the analyses in this study.(2) Previously published reports of sleep metrics in both Drosophila larvae and adults include the average number of sleep episodes (bout number) and the average length of sleep episodes (bout length). Neither of these metrics is included in the paper for either the larval sleep or adult sleep data. Not including these metrics makes it difficult for readers to compare the findings in this study to previously published papers in the established Drosophila sleep literature.(3) Because Drosophila adult & larval sleep is based on locomotion, the authors need to show the activity values for the experiments supporting their key conclusions. They do show travel distances in Figure 2 - Figure Supplement 1, however, it is not clear how these distances were calculated or how the distances relate to the overall activity of individual larvae during sleep experiments. It is also concerning that inactivation of the PK2-R1-expressing neurons causes a reduction in locomotion speed. This could partially explain the increase in sleep that they observe.(4) The authors rely on homozygous mutant larvae and adult flies to support many of their conclusions. They also rely on Gal4 lines with fairly broad expression in the Drosophila brain to support their conclusions. Adding more precise tissue-specific manipulations, including thermogenetic activation and inhibition of smaller populations of neurons in the study would be needed to increase confidence in the presented results. Similarly, demonstrating that larval development and feeding are not affected by the broad manipulations would strengthen the conclusions.(5) Many of the experiments presented in this study would benefit from genetic and temperature controls. These controls would increase confidence in the presented results.(6) The authors claim that their findings in larvae uncover the circuit basis for larval sleep regulation. However, there is very little comparison to published studies demonstrating that neuropeptides like Dh44 regulate larval sleep. Because hugin-expressing neurons have been shown to be downstream of Dh44 neurons, the authors need to include this as part of their discussion. The authors also do not explain why other neuropeptides in the initial screen are not pursued in the study. Given the effect that these manipulations have on larval sleep in their initial screen, it seems likely that other neuropeptidergic circuits regulate larval sleep.

We thank Reviewer #1 for the constructive comments. According to the suggestions, we have compared the relative sleep amounts of wild-type control and Hugin/PK2-R1/IPCs mutants/manipulations between 6hr-period and 18-hour periods in the 2nd instar larval stage and found consistent sleep phenotypes. We have also showed the sleep metrics data of larva and adults. We have included additional data of locomotion and feeding behavior in wild-type control and Hugin/PK2-R1/IPCs mutants/manipulations, which suggest that sleep phenotypes of Hugin/PK2-R1/IPCs mutants/manipulations are less affected by locomotion and feeding behavior changes. As pointed out, our study could not exclude the possibility that in addition to the Hugin/PK2-R1/IPCs axis, other pathways including DH44 could act in larval sleep control. We have included these points in Discussion. Please see point-to-point responses for details.

**Reviewer #2 (Public review):**
Summary:This study examines larval sleep patterns and compares them to sleep regulation in adult flies. The authors demonstrate hallmark sleep characteristics in larvae, including sleep rebound and increased arousal thresholds. Through genetic and behavioral analyses, they identify PK2-R1 as a key receptor involved in sleep modulation, likely via the HuginPC-IPC signaling pathway. Loss of PK2-R1 results in increased sleep, which aligns with previous findings in hugin knockout mutants. While the study presents significant contributions to the field, further investigation is needed to address discrepancies with earlier research and strengthen mechanistic claims.Strengths:(1) The study explores a relatively understudied aspect of sleep regulation, focusing on larval development.(2) The use of an automated behavioral measurement system ensures precise quantification of sleep patterns.(3) The findings provide strong genetic and behavioral evidence supporting the role of the HuginPC-IPC pathway in sleep regulation.(4) The study has broader implications for understanding the evolution and functional divergence of sleep circuits.Weaknesses:(1) The manuscript does not sufficiently discuss previous studies, particularly concerning hugin mutants and their metabolic effects.(2) The specificity of IPC secretion mechanisms is unclear, particularly regarding potential indirect effects on Dilp2.(3) Alternative circuits, such as the HuginPC-DH44 pathway, require further consideration.(4) Functional connectivity between HuginPC neurons and IPCs is not directly validated.(5) Developmental differences in sleep regulatory mechanisms are not thoroughly examined.

We thank Reviewer #2 for the positive comments. As suggested, our study could not exclude the possibility that in addition to the Hugin/PK2-R1/IPCs axis, alternative pathways including the Hugin/DH44 axis could contribute to sleep control in larvae. We have added these points in Discussion. We also have added additional data to show mechanistic differences of larval and adult sleep control. Please see point-to-point responses for details.

**Reviewer #3 (Public review):**
Summary:Sleep affects cognition and metabolism, evolving throughout development. In mammals, infants have fast sleep-wake cycles that stabilize in adults via circadian regulation. In this study, the author performed a genetic screen for neurotransmitters/peptides regulating sleep and identified the neuropeptide Hugin and its receptor PK2-R1 as essential components for sleep in Drosophila larvae. They showed that IPCs express Pk2-R1 and silencing IPCs resulted in a significant increase in the sleep amount, which was consistent with the effect they observed in PK2-R1 knock-out mutants. They also showed that Hugin peptides, secreted by a subset of Hugin neurons (Hug-PC), activate IPCs through the PK2-R1 receptor. This activation prompts IPCs to release insulin-like peptides (Dilps), which are implicated in the modulation of sleep. They showed that Hugin peptides induce a PK2-R1 dependent calcium (Ca²⁺) increase in IPCs, which they linked to the release of Dilp3, showing a connection between Hugin signaling to IPCs, Dilp3 release, and sleep regulation. Additionally, the activation of Hug-PC neurons reduced sleep amounts, while silencing them had the opposite effect. In contrast to the larval stage, the Hugin/PK2-R1 axis was not critical for sleep regulation in Drosophila adults, suggesting that this neuropeptidergic circuitry has divergent roles in sleep regulation across different stages of development.Strengths:This study used an updated system for sleep quantification in Drosophila larvae, and this method allowed precise measurement of larval sleep patterns which is essential for the understanding of sleep regulation.The authors performed unbiased genetics screening and successfully identified novel regulators for larval sleep, Hugin and its receptor PK2-R1, making a substantial contribution to the understanding of neuropeptidergic control of sleep regulation.They clearly demonstrated the mechanism by which Hugin-expressing neurons influence sleep through the activation of IPCs via PK2-R1 with Ca2+ responses and can modulate sleep.Based on the demonstrated activation of PK2-R1 by the human Hugin orthologue Neuromedin U, research on human sleep disorders may benefit from the discoveries from Drosophila since sleep-regulating mechanisms are conserved across species.Weaknesses:The study primarily focused on sleep regulation in Drosophila larvae, showing that the Hugin/PK2-R1 axis is critical for larval sleep but not necessary for adult sleep. The effects of the Hugin axis in the adult are, however, incompletely explained and somewhat inconsistent. PK2-R1 knockout adults also display increased sleep, as does HugPC silencing, at least for daytime sleep. The difference lies in Dilp3/5 mutant animals showing decreased sleep and IPCs seemingly responding with reduced Dilp3 release to PK-2 treatment (Figure 6). It seems difficult to reconcile the author's conclusions regarding this point without additional data. It could be argued that PK2-R1 still regulates adult sleep, but not via Hugin and IPCs/Dilps.Another issue might be that the authors show relative sleep levels for adults using Trikinetics monitoring. From the methods, it is not clear if the authors backcrossed their line to an isogenic wild-type background to normalize for line-specific effects on sleep. Thus, it is likely that each line has differences in total sleep time due to background effects, e.g., their Kir2.1 control line showed reduced sleep relative to the compared genotypes. This might limit the conclusions on the role of Hugin/PK2-R1 on adult sleep.

We thank Reviewer #3 for the valuable comments. According to the suggestions, we have included additional data of adult sleep phenotypes with IPCs/Dilps and HugPC/PK-2 manipulations. We believe that these additional data further support the idea that the Hugin/PR2/IPCs axis acts differently in larval and adult sleep control.

**Recommendations for the authors**:
**Reviewer #1 (Recommendations for the authors):**
(1) Show all data as individual data points in the graphs. The use of box-and-whisker plots makes it difficult to determine how much variation there is in each experiment.

According to the comments, we have changed all graphs to the dots-and-whisker plots (Figures 1–6; Figure 1—figure supplements 2–4; Figure 2—figure supplement 1; Figure 3—figure supplement 1 and 3; Figure 5—figure supplement 1; and Figure 6— figure supplements 1 and 3).

(2) Show all larval sleep metrics (total sleep duration, bout #, bout length, & activity) over the first 6-hour period of 2nd instar development. Larval sleep changes over the course of 2nd instar development so showing an 18-hour period is not as informative for the different manipulations in the study. This also allows for a more thorough comparison to Szuperak et al 2018.

According to the comments, we have shown all larval sleep metrics (total sleep duration, bout #, bout length, & activity) over the first 6 hours for PK2-R1 KO mutants (Figure 1-figure supplemental 5). These PK2-R1 mutant phenotypes are consistent with those described by our sleep amount data over an 18 hr period (Figure 1-figure supplemental 5). We thus consistently show all the sleep phenotype data in the 18 hr period window in the 2nd instar larvae in this paper.

(3) Show activity values for every experiment. Behavior is based on locomotion, so there is a need to show that larvae in each manipulation do not have locomotive defects.

According to the reviewer’s comments, we have shown the activity values for each experiment (Figure 2—figure supplement 1 and Figure 3—figure supplement 1). These data clearly indicated that changes in sleep amounts in each manipulation are not only due to locomotion alterations. We have thus added the sentence below at line 151156 in the manuscript.

Locomotion changes were not consistently observed upon either activation or suppression of Hug neurons (Figure 3—figure supplement 1), suggesting that changes in sleep amounts is unrelated to locomotor alterations.

(4) Provide additional explanation as to why PK2-R1 was pursued in the study. There are several candidates in Figure 1 - Figure Supplement 4 (like sNPF-Gal4, Dh31-Gal4, and DskGal4) that have effects on sleep. These have also not been studied in the context of larval sleep regulation.

According to the reviewer’s comments, we have added the following sentences at line 108-114 in the manuscript.

The role of PK2-R1 in larval sleep, on the other hand, has been unknown to date. Given its strong expression in insulin-producing cells (Schlegel et al., 2016) and its function as a receptor for the neuropeptide Hugin, which modulates feeding (Schoofs et al., 2014), we hypothesized that PK2-R1 might mediate neuropeptidergic signaling that links metabolic and sleep regulation during development. We thus focused on this gene as a candidate connecting behavioral and endocrine sleep control.

(5) Insulin manipulations are known to disrupt Drosophila development (Rulifson et al, 2002). Therefore, it would be beneficial to show that larvae develop normally in dilp3 and dilp5 mutants by examining the time to pupal formation in these mutants compared to controls. If the mutant larvae take longer to reach the pupal stage, how do the authors know that the 2nd instar control and mutant larvae are the same developmental age? As indicated above, the developmental age of larvae does affect the total amount of sleep, so this could affect the authors' conclusions.

We agree that this is an important point in this study. In each experiment, we carefully checked the developmental stage of larvae progeny by mouth hook analysis and measuring larval size and used only larvae with characteristics comparable to wildtype 2nd instar larvae. We have added these descriptions in Methods (line 411–416).

(6) Figure 1 data is only supported by homozygous mutants & 1 fairly-broadly expressed Gal4 driver. The authors need to show that inactivation of PK2-R1 neurons with more tissuerestrictive Gal4 driver lines has the same effect as the other manipulations to further support the conclusions. Examining sleep in activation of PK2-R1 neurons with the broadly expressed Gal4 driver & UAS-TrpA1 would also provide better support for the conclusions.

We agree. Indeed, we tried to narrow down to small subsets of neurons using multiple different Gal4 drivers, but unfortunately, we did not obtain potential candidates.

Therefore, although our data show that the Hugin/PK2-R1axis contributes to sleep control in larvae, we cannot rule out the possibility that other axises could also function in larval sleep control. We mentioned this point in the original version of the submitted manuscript (line 134-137).

(7) Provide more explanation as to how your methods of defining sleep compare/contrast to published papers. It is not clear how many frames = 1 sec in your recordings. The definition of sleep as 12 frames needs to include a time component as well. This allows for easier comparison to other published papers examining Drosophila larval sleep (Szuperak et al 2018; Churgin et al 2019; Poe et al 2023; Poe et al 2024).

Our recordings were acquired at 0.87 frames per second. We have added this information in Method (line 431).

(8) Figure 2 data is only supported by mutants & inactivation with 1 Gal4 driver per cell population. Showing activation of Gal4-expressing cells with UAS-TrpA1 would add more support to the conclusions.

We have already showed the reduced sleep amounts in both HuginGAL4>ReaChR and HuginGAL4>TrpA larvae (Figure 3 C & D) in the original version.

(9) Need to clarify in the methods how the authors calculated travel distances as a measure of locomotive activity. It's not clear if this is done during larval sleep experiments or in independent experiments. It is also not clear why the y-axes of Figure 2-Figure Supplement 1 are not consistent across the panels. Finally, the authors do see decreases in locomotive activity in PK2-R1>Kir2.1 and in dilp3 mutants, so the conclusions presented in the results section of the paper need to be modified to reflect those results.

We calculated travel distances from the same video recording datasets used for sleep quantification. We have added this information in Method (line 431-435). As the reviewer indicated, locomotor activity was reduced in a part of conditions/mutants including PK2-R1 > Kir2.1 and dilp3 mutants, and therefore we cannot exclude the possibility that locomotion changes might contribute to sleep phenotypes. On the other hand, a large part of manipulations of Hugin neurons and IPCs caused a sleep increase without significant changes in locomotor activity (Figure 2—figure supplement 1 and Figure 3—figure supplement 1). It is thus likely that Hugin and IPCs contribute to sleep control independent of locomotion, whereas other neurons trapped by PK2-R1 GAL4 might contribute to locomotion control.

(10) Given the role that hugin neurons play in Drosophila feeding (Schlegel et al, 2016), the authors should include feeding data for the hugin/PK2-R1 manipulations. It is also unclear from the methods if their thresholding for defining sleep can detect feeding behaviors. Changes in feeding behavior could explain some of the reported increases in sleep if feeding is not classified as a waking but is instead picked up as inactivity.

We agree that this is an important point. According to reviewer’s points, we have added feeding amounts of the wild-type control and the HuginPC>Kir2.1 larvae (Figure 3-figure supplement 3). These data suggest that feeding amounts of the HuginPC>Kir2.1 larvae are significantly reduced compared to those of the control. Given that our data analysis typically categorized feeding behavior into “moving (not sleep)” (see Materials and Methods) and that HuginPC>Kir2.1 larvae showed increased sleep amounts compared to the wild-type control, it is likely that the increased sleep amounts in HuginPC>Kir2.1 larvae are unrelated to changes in feeding behavior.

(11) The Hugin-IPC localization data (Figure 3E) would be better supported by the use of more specific synaptic and dendritic markers. Specifically, expressing Syt-eGFP (axon marker) in hugin neurons & DenMark (dendritic marker) in IPCs. Using GRASP or P2X2 to demonstrate the anatomical/functional connections between hugin & IPC neurons would also provide better support for this conclusion.

According to the reviewer’s suggestion, we have added Syt-eGFP signals in HuginPC neurons (Figure 4—figure supplement 1). We also tried DenMark expression in IPCs, but we could not obtain dipl3>DenMark F1 progeny for unknown season. We also applied GRASP to the HuginPC-IPCs interaction, but we could not detect obvious GRASP signals. It is well known that peptidergic transmission is often independent of conventional synapse structures, called as volume transmission, in which peptidergic signals can transmit over a long-range distance to targeting neurons. It is thus possible that IPCs might receive Hugin signals from HuginPC neurons through volume transmission.

(12) Figure 4 is missing temperature controls for thermal activation experiments. Also missinggenetic control for UAS/+. It would be more convincing to see experiments in Figure 4 with the more specific hug-PC-Gal4 line instead of the broadly expressed hugin-Gal4 line.

According to reviewer’s comments, we have added the control data in Figure 4.

(13) Representative images for Figure 4B & 4C would provide better support for the quantifications & conclusions presented.

According to the reviewer’s suggestions, we show the representative imagine for Figure 4B and 4C (please see Author response image 1). We are, however, afraid that these images might not help readers’ further understanding in addition to the quantitative data, so we have decided to not add these images in the manuscript.

**Author response image 1. sa3fig1:** mCD8::mCherry (top) and CRTC::GFP (bottom) are shown under high-temperature conditions without ("−") or with ("+") hugin neuron activation. "-" denotes a high-temperature genetic control lacking LexAop-TrpA1, thus no thermogenetic activation occurs. CRTC::GFP is shown in pseudocolor.

(14) A more zoomed-out image of all the IPC neurons in the bath application of hugin peptides (Figure 5D) would help with the interpretation of the results. It's not clear if the authors only measured the same exact neuron in each IPC cluster or if they examined all of the IPC neurons. If they measured all of the IPC neurons, did they observe similar results across the different neurons? How much variability is there in the response of IPC neurons to hugin peptide application?

For Figure 5, we obtained images of multiple brains from each genotype and quantified the NLI values from all IPC neurons. For reference, we show plots of the CRTC signals of Figure 5C each brain by bran (Author response image 2). We have added detailed information of CRTC analysis in Methods (lines 552-554).

**Author response image 2. sa3fig2:** Distribution of CRTC signals across individual brains. Plots of nuclear localization index (NLI) for individual brains, corresponding to the conditions shown in Figure 5C. The x-axis represents each larval brain preparation, and each dot indicates the NLI value of a single IPC neuron. Horizontal bars represent the median within each brain. These plots illustrate variability both within and across individual brains.

(15) The conclusion that application of Hug peptides results in dilp3 release is not well supported (Figure 5E). There is a large amount of variation in anti-dilp3 signal. Representative images for these quantifications would be beneficial. The authors also don't directly show that dilp3 vesicles are released. They only see a reduction in antibody accumulation in IPCs. Could there be other reasons for the reduction in accumulation in the IPCs? Would changes in dilp3 gene expression or membrane localization cause a reduction in signal? Showing that actual release of dilp3 is affected by Hug peptides using a reporter like ANF-GFP would be more convincing.

According to the reviewer’s comments, we have added representative images (Figure 5—figure supplement 2). As for the ex vivo experiments in Fig5, we treated the extracted brain tissues with Hugin/NMU peptides for only 5minutes. It is thus most likely that reduction of Dilps in IPCs is mediated by Hugin/PK2-R1 signal-dependent secretion, rather than transcriptional control and/or degradation of Dilps.

(16) Show all sleep metrics (total sleep duration, bout #, bout length, and activity) for adult sleep experiments. Showing relative total sleep for the adult experiments is confusing & would benefit from plots of total average sleep in minutes for each genotype.

According to the reviewer’s comments, we have added the sleep metrics in adults (Figure 6; Figure 6-figure supplement 3).

(17) The authors can't conclude that expression patterns of PK2-R1 & hug between larvae & adults are "almost comparable." They don't track neurons over development or immortalize neurons in larvae & check expression patterns in adults. They need to show some type of quantification to support these claims. Or revise the text to remove this conclusion.

We agree. We have changed our augments as follow (line 211-214).

Interestingly, the expression patterns of PK2-R1 and Hug as well as the morphology of HugPC neurons in adults appeared to be similar to those in larvae (Figure 6—figure supplement 2), implying that the differential roles of Hug in larvae vs adults are likely due to physiological differences in HugPC neurons and/or IPCs.

(18) For Figure 6, what effect does genetic inactivation of IPCs have on adult sleep? A more specific manipulation of these cells would provide better support for the conclusion that IPC manipulations have distinct effects on larval & adult sleep. The sleep traces for the hugin manipulation & dilp mutants (Figure 6-Figure Supplement 1) also look inconsistent when comparing genetic controls in (Figure 6-Figure Supplement 1D) or when comparing the dilp mutants. Plotting this data as total sleep amount in the day & night (2 separate graphs) would be beneficial. It would also be helpful to see additional sleep traces for these experiments.

According to the reviewer’s comments, we have added the sleep amounts of added *dilp3* and *dilp5* adults (Figure 6-figure supplement 1C-D) as well as IPC silencing (Figure6-figure supplement 3D) in a daytime/night time sleep-separated manner.

(19) For Figure 6, what effect does thermogenetic activation of hugin neurons have on IPC activity? The authors demonstrate in Figure 5 that thermal activation results in an increase in larval IPC activity, but they do not show these experiments in the adult brain. These experiments would provide more support for their conclusion that hugin has differential effects on IPC activity depending on the developmental age (larvae vs adults).

According to the reviewer’s comments, we performed thermo-activation of hugin neurons and found no significant effects on adult IPCs (see Author response image 3), consists with the ex *vivo* data in Figure 6.

**Author response image 3. sa3fig3:** 

(20) A figure legend is needed for Figure 7. The model is not self-explanatory, nor is there an adequate explanation in the discussion section.

We have added legends (line 781-785).

(21) Since hugin is known to be downstream of Dh44 in larvae, the discussion needs to include comparison to published work on Dh44 in larvae (Poe et al, 2023). The hugin receptor, PK2R1, is also expressed in Dh44 & DMS neurons (Schlegel et al, 2016), so a discussion of what role Dh44/DMS neurons may play in their model is necessary.

We agree. We have added discussion as follow in Discussion (line 313-320).

We cannot rule out the possibility that other neurons could function downstream of HuginPC neurons in sleep regulation. For instance, given that Dh44 neurons in the brain promote arousal (Poe et al. 2023) and are PK2-R1-positive (Schlegel et al. 2016), Hugin might control sleep in part through Dh44 neurons.

(22) Minor point: Line 97 should say "resulted in a significant sleep increase." Currently, it says "decrease" which is not what is depicted in the figure.

We appreciate the reviewer’s point. We have corrected this.

(23) Minor point: Figure 5 should be renamed as Figure 4 since the text describing the results in Figure 5A & 5B occurs before the text describing the results in Figure 4.

We do understand the point the reviewer arose. However, since Fig5A explains the experimental setup of the whole Fig5s, we would like to keep Fig5A at the original position.

**Reviewer #2 (Recommendations for the authors):**
First, the study would benefit from a more comprehensive discussion of previous research, particularly the work by Schlegel et al. (2016) and Melcher and Pankratz (2006). A key inconsistency that should be addressed is the observation that hugin mutant larvae exhibit reduced body size and feeding behavior, which may influence Dilp2 secretion. The selective effect on Dilp3 and Dilp5 without affecting Dilp2 warrants further clarification. Conducting conditional gene expression experiments to control hugin, dilp3, and dilp5 expression, along with neuronal activity modulation, would help determine whether the observed effects are direct or secondary consequences.

According to the review’s comments, we tried to manipulate neuronal activity in IPCs, but unfortunately, expression of Kir2.1 in IPCs caused die or very weak animals. Instead, we cited a recent paper that shows a differential secretion of Dilp2 and Dilp6 in a stimulant-dependent manner (Suzawa et al. PNAS 2025) and added more discussion about selective Dilp3/5 secretion by Hugin-PK2-R1 signals (line 275-297).

Second, the specificity of IPC secretion mechanisms should be clarified. Given that IPCs coexpress Dilp2, Dilp3, and Dilp5, it remains unclear how the pathway selectively modulates Dilp3 and Dilp5 while leaving Dilp2 unaffected. Additional experiments, such as electron microscopy, could provide insights into whether anatomical differences in vesicular pools influence peptide secretion. Since hugin mutants are reported to have reduced body size, confirming that Dilp2 secretion remains truly unchanged is crucial for eliminating potential indirect effects.

We thank this reviewer for the valuable suggestions. Since the selective Dilp secretion mechanisms in IPCs are not the main scope in this paper, we would like to attempt detailed EM analysis in next studies. We cited a recent paper that shows a differential secretion of Dilp2 and Dilp6 from IPCs in a stimulant-dependent manner (Suzawa et al. PNAS 2025) and added more discussion about selective Dilp3/5 secretion by Hugin-PK2-R1 signals (line 275-297).

Third, the study should explore the potential role of alternative circuits, such as the HuginPCDH44 pathway, in sleep regulation. The observation that DH44 mutants exhibit even greater sleep amounts than PK2-R1 mutants suggests the involvement of additional regulatory mechanisms. Prior studies indicate that HuginPC neurons may influence DH44 neuron activity, which could impact sleep. Furthermore, recent findings link DH44 with starvation-induced sleep loss in adult flies. Discussing and experimentally investigating the HuginPC-DH44 axis in larval sleep regulation would provide additional depth to the study.

As far as we understand, any direct evidence for HuginPC→DH44 pathway has not been reported in larvae as well as adults. Instead, DH44 influences Hugin neuron activity in adults (King et al. 2017). We thus examined whether optogenetic DH44 activation could influence HuginPC activity using CRTC analysis, but unfortunately, we could not detect significant changes in HuginPC activity.

Given that PK2-R1 is expressed in DH44-positive neurons (Schelgel et al 2016) and that DH44-positive neurons are localized at the regions to which HuginPC neurons innervate, it is still possible that the HuginPC→DH44 pathway might function in parallel to the HuginPC→IPCs pathway. We feel that this is quite an interesting possibility and should be a nice scope in the next paper.

Fourth, validating the functional connectivity between HuginPC neurons and IPCs using calcium imaging would significantly enhance the study. Employing real-time calcium imaging with GCaMPs would provide direct evidence of synaptic activity between these neuronal populations. Such data would strengthen the claim that the observed sleep regulatory effects result from direct neural communication rather than secondary systemic influences.

We agree. Indeed, we tried Ca^2+^ imaging of HuginPC neurons and IPCs in living larvae as well as using ex vivo preparations, and realized that it was quite technically difficult to obtain reliable Ca^2+^ dynamics data in the brain of living larvae/ex vivo brain tissue. Therefore, instead of live Ca^2+^ imaging, we performed the CRTC analysis using fixed brain preparations. We have added the mention that we tried Ca^2+^ imaging in the larval brain, but it did not work well (line 555-558).

Finally, a more detailed discussion of developmental differences in sleep regulatory mechanisms would be beneficial. The manuscript should address why genes involved in sleep modulation during development may function differently from their roles in adult sleep regulation. Providing a conceptual framework or experimental evidence to explain these developmental differences would enhance the study's contribution to understanding the evolution of sleep circuits. Clarifying how these findings fit into broader sleep regulation models would increase the impact of the research.

We agree. We would like to add discussions about how factors/circuits involved in sleep modulation during development may function differently from their roles in adult sleep regulation as follows (line 349-371), as it is rather difficult to discuss why.

It is thus possible that Hugin/PK2-R1 signaling along the HugPC-IPCs circuitry is suppressed in adults. IPCs in adults receive multiple positive and negative modulatory inputs through GPCRs including the metabotropic GABA_B_ receptors (Enell et al., 2010), which suppresses IPC activity and Dilp release in adult IPCs (Enell et al., 2010). It is thus plausible that such negative modulatory inputs to IPCs in adults might counteract with the Hugin/PK2-R1 axis to suppress Dilp release. In addition, our data suggest that Dilps modulate sleep amount in the opposite directions in larvae and adults (Figure 7). Comparing the expression levels and activities of GPCRs in larval and adult IPCs would be essential to better understand how the same modulatory signals over the course of development come to exert differential impacts on sleep. Interestingly, Hugin in adults appears irrelevant for the baseline sleep amount but is required for homeostatic regulation of sleep (Schwarz et al., 2021). Thus, testing if Hugin/PK2-R1 axis is involved in the homeostatic regulation of larval sleep, and how such a system compares to its adult counterpart, may further provide mechanistic insights into how homeostatic sleep regulation matures over development.

By addressing these aspects, the manuscript will provide a clearer, more robust, and wellsupported analysis of larval sleep regulation. These refinements will help improve the study's clarity and impact, ensuring that its findings are effectively communicated to the research community.
**Reviewer #3 (Recommendations for the authors):**
(1) Line 97: "Silencing neurons expressing Oamb and PK2-R1 resulted in a significant sleep decrease?" But there is an increase in sleep amounts from Figure 1A. (Typo error).

We thank the reviewer for pointing out this typo. We have corrected this typo in the revised version.

(2) Line139: "HugPC and IPCs labeled by Dilp3-GAL4 are located in close proximity to each other." While proximity does not equal synaptic connections, direct connectivity of HugPC and IPCs was already shown in larval connectome analyses with HugPC providing the strongest input of larval IPCs (Hückesfeld et al. eLife 2021). This could be cited in this context instead.

We agree. We have cited this paper in References (line 163).

(3) Figure 2 Supplement 1: Locomotion speed is affected in PK2-R1 knockouts; what is the significance regarding the observed sleep increase?

We agree that this is a very important point. As the reviewer pointed out, since locomotion speed was reduced in PK2-R1 KO larvae, sleep increase phenotype in PK2-R1 KO larvae might be in part due to reduction of locomotion. On the other hand, IPCs silencing by Kir2.1caused sleep increase phenotype without significant changes in locomotion (Figure 2; Figure 2-figure supplement 1). It is thus possible that since PK2-R1 is broadly expressed in the nervous system in addition to IPCs (Figure 2), PK2-R1 neurons other than IPCs might contribute to locomotion control.

(4) Why are Dilp3 levels changing (increasing) in adult IPCs after PK-2 treatment? This is not mentioned in the text and is not discussed at all.

As the reviewer indicated, this data is unexpected to us. At this moment, we could only assume that PK-2 could act in larval and adult IPCs in a different manner. We have added this sentence in Results (line 211-214).

(5) It has been shown in other publications that Dilps play a role in sleep regulation (Cong et al., Sleep 2015), this study should be cited.

We have cited this paper in References (line 224).

(6) The order of discussing figure panels is sometimes confusing, e.g. Figure 6C is discussed at the very end after 6D-F.

We agree. Indeed, we discussed a lot about this order during preparation of the first draft. However, we finally decided the current order, as grouping “sleep phenotype data” and “ex vivo data” should be easier to understand for readers. We thus keep the current order in the revised submission.